# It's All Just Vectorization: einx, a Universal Notation for Tensor Operations

**Florian Fervers   Sebastian Bullinger   Christoph Bodensteiner   Michael Arens**
Fraunhofer IOSB
`{firstname.lastname}@iosb.fraunhofer.de`

## Abstract

Tensor operations represent a cornerstone of modern scientific computing. However, the Numpy-like notation adopted by predominant tensor frameworks is often difficult to read and write and prone to so-called shape errors, *i.a.*, due to following inconsistent rules across a large, complex collection of operations. Alternatives like einsum and einops have gained popularity, but are inherently restricted to few operations and lack the generality required for a universal model of tensor programming.

To derive a better paradigm, we revisit vectorization as a function for transforming tensor operations, and use it to both lift lower-order operations to higher-order operations, and conceptually decompose higher-order operations to lower-order operations and their vectorization.

Building on the universal nature of vectorization, we introduce *einx*, a universal notation for tensor operations. It uses declarative, pointful expressions that are defined by analogy with loop notation and represent the vectorization of tensor operations. The notation reduces the large APIs of existing frameworks to a small set of elementary operations, applies consistent rules across all operations, and enables a clean, readable and writable representation in code. We provide an implementation of einx that is embedded in Python and integrates seamlessly with existing tensor frameworks: `https://github.com/fferflo/einx`

## 1 Introduction

Tensor operations constitute the foundation of modern deep learning and other domains of scientific computing. Tensors, *i.e.* $n$-dimensional arrays with a uniform element type, serve as a medium for diverse types of data, including images, volumes, sequences of audio or text, activations in a neural net, class probability scores, or batches thereof. Tensor programs are commonly written in high-level Python with tensor operations that act as points of entry to low-level backend routines, thereby abstracting from the underlying hardware, memory representation and algorithms.

The widely used *Numpy-like notation* for expressing tensor operations in Python is followed by most predominant tensor frameworks such as Numpy itself (Harris et al., 2020), PyTorch (Paszke, 2019), Tensorflow (Abadi et al., 2015), Jax (Bradbury et al., 2018), and MLX (Hannun et al., 2023). An operation in Numpy-like notation operates on whole tensors and is expressed, *e.g.*, as follows:

```python
y = np.sum(x, axis=1) # Compute sum along rows of the matrix x
```

In contrast, the following representation of the same operation in *loop notation* addresses tensor elements individually using indices, and invokes a backend routine multiple times:

```python
for i in range(x.shape[0]):
    for j in range(x.shape[1]):
        y[i] += x[i, j] # For each row i, add element from column j
```

Loop notation enables a clearer, more general representation of tensor operations through its *pointful* style, *i.e.* explicit use of indices. In contrast, Numpy-like notation follows a *point-free* style (Paszke et al., 2021) and compensates for the lack of index expressions by introducing varying mechanisms, including special parameters (*e.g.*, `axis`), pure shape operations, as well as broadcasting, advanced indexing, and numerous function-specific rules. This often results in tensor programs that are difficult to read and write and where so-called shape errors occur frequently.

Table 1: **It's all just vectorization:** einx reduces the large, inconsistent API of Numpy-like frameworks to few elementary operations and a universal, declarative, pointful notation for expressing their vectorization. The table shows examples of different Numpy-like function calls that map to the same elementary operation in einx and differ solely in their vectorization.

| Numpy-like notation | einx notation (ours) |
|---|---|
| `torch.take(x, y)` | `einx.get_at("[x], ... -> ...", x, y)` |
| `torch.gather(x, 0, y)`
`torch.take_along_dim(x, y, dim=0)` | `einx.get_at("[x] b c, i b c -> i b c", x, y)` |
| `torch.index_select(x, 1, y)`
`tf.gather(x, y, axis=1)` | `einx.get_at("a [x] c, i -> a i c", x, y)` |
| `tf.gather_nd(x, y)` | `einx.get_at("[...], b [i] -> b", x, y)` |
| `tf.gather_nd(x, y, batch_dims=1)` | `einx.get_at("a [...], a b [i] -> a b", x, y)` |
| `x[y[:, 0], y[:, 1]]` | `einx.get_at("[x y], a [2] -> a", x, y)` |
| `x * y[:, np.newaxis]` | `einx.multiply("a b, a -> a b", x, y)` |
| `np.outer(x, y)` | `einx.multiply("a, b -> a b", x, y)` |
| `np.kron(x, y)` | `einx.multiply("a..., b... -> (a b)...", x, y)` |
| `scipy.linalg.khatri_rao(x, y)` | `einx.multiply("a c, b c -> (a b) c", x, y)` |
| `np.matmul(x, y)` | `einx.dot("a [b], [b] c -> a c", x, y)` |
| `np.dot(x, y)` | `einx.dot("x [a], y [a] b -> x y b", x, y)` |
| `np.tensordot(x, y, axes=(0, 1))` | `einx.dot("[a] b, c [a] -> b c", x, y)` |
| `np.inner(x, y)` | `einx.dot("x [a], y [a] -> x y", x, y)` |
| `np.transpose(x, (0, 2, 1))` | `einx.id("a b c -> a c b", x)` |
| `np.squeeze(x, axis=1)` | `einx.id("a 1 c -> a c", x)` |
| `np.expand_dims(x, axis=1)` | `einx.id("a c -> a 1 c", x)` |
| `np.broadcast_to(x, (2, 3, 4))` | `einx.id("c -> 2 3 c", x)` |
| `np.reshape(x, (-1,))` | `einx.id("... -> (...)", x)` |
| `np.concatenate([x, y], axis=-1)` | `einx.id("s a, s b -> s (a + b)", x, y)` |
| `np.stack([x, y], axis=0)` | `einx.id("..., ... -> (1 + 1) ...", x, y)` |
| `np.meshgrid(x, y, indexing="ij")` | `einx.id("a, b -> a b, a b", x, y)` |

Several alternatives to Numpy-like notation have been proposed, including approaches inspired by Einstein's summation convention such as einsum (Wiebe, 2011) and its extension einops (Rogozhnikov, 2022a), frameworks that shift from positional to symbolic dimensions (Hoyer & Hamman, 2017; DeVito, 2023), and custom pointful languages (Vasilache et al., 2018; Paszke et al., 2021). Of these, only einsum and einops have found widespread adoption in the deep learning community, *i.a.*, due to being embedded in Python and compatible with the existing Numpy-based ecosystem.

However, einsum and einops are inherently restricted to a limited set of operations (*c.f.* Tab. 2) and lack the generality required for a universal model of tensor operations. Furthermore, einops is defined in large parts ostensively, *i.e.* by examples such as

```
y = einops.reduce(x, "a b -> a", reduction="sum") # Sum-reduction along b
```

rather than by a clear, explicit interpretation of how terms such as `"a b -> a"` are to be understood.

**Our contributions are as follows:**

(1) We revisit vectorization as a general function for transforming tensor operations. We use it as a universal tool to lift lower-order operations to higher-order operations, and conceptually decompose existing higher-order operations to few lower-order operations and their varying vectorization.

(2) We introduce a universal notation for tensor operations: *einx*. It represents the vectorization of operations using declarative, pointful expressions that are defined by analogy with loop notation. The einx notation (a) is applicable to *any* tensor operation, (b) provides a *single* set of rules across arbitrary operations, (c) is interpretable by analogy with loop notation, (d) allows for a clean, readable and writable representation of operations in code, and (e) reduces the complex Application Programming Interface (API) of Numpy-like frameworks to few elementary operations (*c.f.* Tab. 1).

(3) We provide an implementation of einx for widely used tensor frameworks. Operations in einx are compiled to function calls of a given tensor framework and thereby allow for a seamless integration. The einx API contains functions for many commonly used operations, and the option to adapt new, custom operations to einx notation.

Table 2: Support for classes of operations and vectorization in different types of ein*-notation. P: Permutation. F: Flattening. R: Repetition (*i.e.* output-only vectorization). C: Concatenation. *: Always and only flattens concatenated axes. **: Coordinate axis must be first axis.

| Operation | einx (ours) | einsum (2011) | einops: reduce, repeat, rearrange, einsum (2022a) | einops: pack, unpack (2022b) | eindex (2023) |
|---|---|---|---|---|---|
| Identity | PFRC | P | PFR | (FC)* | - |
| Scalar | PFR | P *(only mul.)* | P *(only mul.)* | - | - |
| Reduction | PFR | P *(only sum)* | PF | - | - |
| Dot-product | PFR | P | P | - | - |
| Indexing | PFR | - | - | - | (P)** |
| Any other | PFR | - | - | - | - |

## 2 RELATED WORKS

### 2.1 EIN*-NOTATIONS FOR TENSOR OPERATIONS

**Einstein Summation**    Einstein (1916) introduces what is now known as the *Einstein summation convention* in the mathematical notation of tensor contractions (*i.e.* generalized matrix multiplications) as follows (translated from German original): *"It is therefore possible, without compromising clarity, to omit the summation signs. To that end, we introduce the rule: If an index appears twice in a term of an expression, it is always to be summed over"*. As an example, in the following contraction of $A$ and $B$ the index $j$ appears twice, and the summation sign over $j$ may therefore be omitted:

$$\sum_j A_{ij} B_{jk} = A_{ij} B_{jk}$$

**einsum**    After early proposals to express Einstein summation in code (Barr, 1991; Åhlander, 2002), the most common approach in Python follows the `np.einsum` function introduced in Numpy by Wiebe (2011). In its rarely used *Einstein mode*, einsum represents a tensor contraction by listing the indices from the corresponding Einstein summation expression in a comma-delimited string:

```
np.einsum("ij,jk", A, B) # Matrix multiplication (as above)
```

Since the index `j` appears twice, it is summed over following Einstein's summation convention.

The function also introduces the more widely used *non-Einstein mode* where the expression is extended using an arrow and output indices as shown below. Instead of Einstein's summation convention, it applies the following rule: All indices that appear only on the left side of the arrow are summed over. This allows expressing additional, commonly used operations, *e.g.*:

```
np.einsum("bij,bjk->bik", x, y) # Batched matmul: j is summed over
np.einsum("ij->i", x) # Sum-reduction: j is summed over
```

Lastly, the ellipsis `...` is used to represent a variable number of indices in an einsum expression.

**einops**    Rogozhnikov (2022a) introduces *einops* which extends the *non-Einstein* mode of einsum to support additional reduction operations using the same notation (*e.g.* max, mean), broadcasting along axes in the output expression, and multi-letter axis names. Its main novelty is the *axis composition* which allows (un)flattening axes by wrapping them in parentheses in the string expression:

```
einops.rearrange(x, "a b c -> (a b) c") # Reshape operation
```

**ein***    Several software packages propose variants of the above notation to support new operations, including *einindex* (Malmaud, 2018), `einops.{pack|unpack}` (Rogozhnikov, 2022b), *eindex* (Rogozhnikov, 2023), *eindex* (McDougall, 2023), *eingather* (Fleuret, 2023) and *einmesh* (Jensen, 2025). However, these variants are incompatible with the original einsum and einops notation as well as with each other, and do not represent a universal notation for tensor operations.

Despite their name, operations in einops and these packages do not apply Einstein's summation convention. Instead, they follow Wiebe's orthogonal choice to use a string of axis names akin to indices in mathematical notation. Since the *ein** terminology has become associated with this notational style, we name our approach *einx*, but avoid claims of it being "Einstein-like" or "Einstein-inspired".

**einx**    We introduce einx, a *universal* notation for tensor operations that follows a single set of notational rules across any given operation (*c.f.* Tab. 2). It is defined by analogy with loop notation, which allows for an explicit interpretation of expressions such as `"i j -> i"`. The notation is compatible with einops notation for the set operations also supported by einops.

## 2.2 OTHER NOTATIONS FOR TENSOR OPERATIONS

**Named tensors**    Several authors (Hoyer & Hamman, 2017; Chen, 2017; Hall et al., 2022; DeVito, 2023; Johnson, 2024) propose to annotate tensors with symbolic axis names, resulting in so-called *named tensors*. Named tensors address the complexity of Numpy-like notation by implicitly vectorizing operations along matching symbolic axes of the argument tensors. However, named tensors also do not self-document tensor shapes, require renaming of axes in tensor programs, and do not integrate seamlessly with the scientific Python ecosystem which operates on positional tensors.

Importantly, the usage of named tensors is complementary to the usage of ein*-notation. Operations may accept named tensors akin to positional tensors by matching the string expression against the symbolic axis names of the tensor, rather than or in addition to the axis positions. This is done, *e.g.*, by the Haliax framework (Hall et al., 2022) which implements einsum for named tensors.

**Other pointful notations**    Several authors (Vasilache et al., 2018; Paszke et al., 2021; Bachurski & Mycroft, 2024) propose other types of pointful notation to express tensor operations using index expressions. They define a set of elementary operations as well as a notation to compose more complex tensor operations. However, there is only limited integration with existing tensor frameworks and support for vectorizing operations that are not defined in the notation itself.

## 2.3 DEFINITION OF VECTORIZATION IN LITERATURE

The term *vectorization* has been used in different contexts within tensor programming. Harris et al. (2020) describe element-wise operations such as np.{add|multiply} in Numpy as *vectorized operations*: These operations apply a scalar function to higher-dimensional tensors in conjunction with broadcasting rules to match axes across arguments. In contrast to our general perspective on vectorization, Harris et al. do not use the term w.r.t. other types of operations such as np.{sum|matmul}.

In named tensor frameworks (*c.f.* Sec. 2.2), lifting of operations to higher-dimensional argument tensors emerges implicitly as a by-product of introducing symbolic axes, and is sometimes referred to as vectorization (Chiang et al., 2023). In this context, Chiang et al. identify some elementary operations that may be vectorized to represent complex functions in Numpy-like frameworks, including scalar, reduction, dot-product, vector-to-vector and indexing operations. However, support and adoption of the notation in existing frameworks remains limited (*c.f.* Appendix H).

Bradbury et al. (2018) introduce jax.vmap (vectorizing map) which regards vectorization as a transformation of operations: Given any operation op, the result of jax.vmap(op, ...) is a new operation that accepts and returns tensors with up to one more dimension than op along which the vectorization is applied. While this allows for a general perspective on vectorization, it does not represent a concise, declarative notation for tensor operations, and is not posed as a universal alternative to Numpy-like notation.

## 3 VECTORIZATION

### 3.1 VECTORIZING LOWER-ORDER OPERATIONS TO HIGHER-ORDER OPERATIONS

We define vectorization as the transformation of an operation that processes a single data point into an operation that processes a collection of data points simultaneously. For instance, the $\sin$ function accepts and returns a scalar, while a vectorized $\sin$ function accepts and returns a collection of scalars and applies the $\sin$ function to each scalar separately. This broad definition of the term differs from its specific use in compiler design where it describes the automatic substitution of scalar instructions with vector instructions following the Single Instruction, Multiple Data (SIMD) model. (Cui, 2024)

In the context of tensor programming, vectorization is applied *along axes* of the tensor arguments: An operation that is vectorized along a particular axis with length $l$ of an $n$-dimensional argument tensor is applied to each of the $l$ separate $(n-1)$-dimensional sub-tensors that are stacked along this axis. For instance, a vectorized $\sin$ function that operates on 1-dimensional vectors with length $l$ computes the $\sin$ of $l$ separate 0-dimensional scalars. Vectorizing an operation is also known as *lifting* the operation to higher-order (*i.e.* higher-dimensional) tensors, or applying the operation to a *batch* of data.

Loop notation provides a natural representation of vectorized operations by expressing the repeated application of the elementary operation to the individual sub-tensors stacked along a given axis. For instance, the following code represents the vectorized sin operation that accepts and returns vectors:

```
for i in range(x.shape[0]):
    y[i] = sin(x[i])
```

The terms x[i] and y[i] represent the scalar sub-tensors that are stacked along the first axis of the vectors x and y and are forwarded to the sin operation. The representation with loop notation is only for conceptual reasons and does not indicate how the operation is actually implemented.

Vectorization along multiple axes is represented using multiple for loops and analogous to multiple consecutive one-dimensional vectorizations along each of the respective axes:

```
for i in range(x.shape[0]): for j in range(x.shape[1]):
    y[i, j] = sin(x[i, j])
```

We consider vectorization only w.r.t. operations that are invariant to the order of the loops and indices per loop, and omit loops in the following examples.

The usage of a subset of the available loop variables to address the axes of a specific tensor expresses what is known as *broadcasting* in Numpy-like notation (Harris et al., 2020):

```
z[i, j] = x[i] * y[j] # Outer product of x and y
```

Lastly, we consider elementary operations that are applied to non-scalar arguments. For instance, softmax operates on vectors and is vectorized along the second dimension of a matrix as follows:

```
y[:, i] = softmax(x[:, i])
```

The terms x[:, i] and y[:, i] represent the one-dimensional sub-tensors that are stacked along the second dimension of the matrices x and y and are forwarded to the softmax operation. We say that the softmax operation is *applied along the first axis* and *vectorized along the second axis* of x and y. We denote axes that the elementary operation is applied along as *argument sub-tensor axes*, and all other axes as *vectorized axes*.

## 3.2 DECOMPOSING HIGHER-ORDER OPERATIONS TO LOWER-ORDER OPERATIONS

In the previous section, we considered the vectorization of lower-order operations to higher-order operations. We now go the opposite direction and conceptually decompose many existing higher-order operations, *e.g.*, from Numpy-like notation, to few lower-order operations and their varying vectorization. In the following, we provide several examples.

A matrix multiplication is represented conceptually as a *vectorized dot-product*, and its inherent vectorization is expressed in loop notation as follows:

```
z[i, j] = dot(x[i, :], y[:, j])
```

Other types of tensor contractions (*e.g.*, np.{dot|matmul|tensordot|inner}) analogously represent vectorized dot-products, but differ in their vectorization. Their implementation typically employs optimized algorithms that do not simply loop over invocations of the elementary dot-product.

The sum-reduction operation np.sum(x, axis=1) over a matrix x is decomposable, *i.a.*, using two alternative choices for the elementary operation:

```
y[i] = sum(x[i, :]) # "sum" maps a vector to a scalar -> 1 vectorized axis
y[i] += x[i, j] # "+=" adds a scalar to a scalar -> 2 vectorized axes
```

Different types of multiplication such as the outer, Hadamard, Kronecker, and Khatri–Rao products are represented as *vectorized scalar multiplication* and differ solely w.r.t. their vectorization.

Shape operations such as np.{transpose|reshape} are represented as *vectorized identity maps*

```
y[j, i] = identity(x[i, j]) # Transpose
y[i * x.shape[1] + j] = identity(x[i, j]) # Reshape/flatten
```

with identity(a) = a. Implementations of these operations typically only modify a tensor's meta-data, rather than applying an assignment or copy operation per element.

Broadcasting tensors along new axes (*e.g.*, np.{broadcast_to|tile|repeat}) is represented as an identity map that is vectorized, *i.a.*, along dimensions which appear only in the output:

```
y[i, j] = identity(x[i]) # Broadcast along j
```

Indexing operations such as `torch.{take|gather|take_along_dim|index_select}` are vectorized versions of the following elementary operation: Retrieve a single value from an $n$-dimensional value tensor at the coordinates specified by a one-dimensional coordinate vector with length $n$. For instance, the following operation gathers color values from an image at the given pixel coordinates:

```
# image: (height, width, #channels)    pixels: (#pixels, 2)
y[p, c] = get_at(image[:, :, c], pixels[p, :])
```

As illustrated above, many tensor operations in Numpy-like frameworks reduce to few elementary operations when factoring out their vectorization. *It's all just vectorization - and always has been!* We use this observation in the following section to define a universal notation that represents tensor operations as vectorized elementary operations.

## 4  EINX

### 4.1  NOTATION

**Overview**   An operation in einx is expressed using the following function call signature:

```
{outputs...} = einx.{elementary_operation}("{vectorization}", {inputs...})
```

This code states that the operation `{elementary_operation}` is vectorized according to the expression `"{vectorization}"`, accepts the tensors `{inputs...}`, and returns the tensors `{outputs...}`. einx provides *one* entry-point per elementary operation and follows Numpy's naming of operations where possible. For instance, the following represents a vectorized scalar addition similar to `np.add`:

```
z = einx.add("{vectorization}", x, y)
```

**Vectorization**   The vectorization string is constructed by analogy with loop notation as follows:

(1) Express the operation in loop notation (*c.f.* Sec. 3). To illustrate this, we consider the following example tensor operation that vectorizes `SOME_OPERATION`:

```
for a in range(...): for b in range(...):
    z[a, :, b] = SOME_OPERATION(x[:, :, a], y[b])
```

The number of colons (`:`) per tensor indicates the dimensionality of the arguments and return values of the elementary operation: Here, the first input is a matrix, the second input a scalar, and the output a vector.

(2) Take the expressions that are used to denote sub-tensors (here: `x[:, :, a]`, `y[b]`, `z[a, :, b]`), and convert the indices to the vectorization string as follows:

  (a) Use an arrow (`->`) to delimit inputs from outputs.
  (b) Use commas to delimit multiple tensors on each side of the arrow.
  (c) Use spaces to delimit indices per tensor.
  (d) Replace colons (`:`) with new symbolic axis names and place brackets (`[]`) around them.

Applying these rules results in the following einx representation for the above example operation:

```
z = einx.SOME_OPERATION("[c d] a, b -> a [e] b", x, y)
```

The vectorization expression indicates the shapes of input and output tensors. Here, x, y, and z have shapes `(c, d, a)`, `(b)`, and `(a, e, b)`, respectively. Unlike in loop notation where index names denote loop variables, in einx notation the symbolic names refer to tensor axes. The loop ranges are determined implicitly from the given tensor dimensions.

Brackets denote axes of the argument sub-tensors that are passed to the elementary operation: `SOME_OPERATION` is invoked with tensors of shapes `(c, d)` and `()`, and returns a vector with shape `(e)`. Brackets may appear both in input and output expressions, and must be placed around the number of axes that matches the dimensionality expected by the elementary operation. Axes not marked with brackets are vectorized. The same axis name may be used for multiple sub-tensor argument axes, *e.g.*, to indicate that they must have the same length:

```
z = einx.dot("a [b], [b] c -> a c", x, y) # Matrix multiplication
```

**Axis composition**   Some tensor operations are representable in loop notation by mapping one or more of the loop variables to a new index value (*e.g.*, `np.reshape` uses the row-major formula). We analogously define the following *axis compositions* in einx notation as ways in which one or more axes are combined to form a single, new axis in the expression.

We define a *flattened axis* as multiple axes of a single tensor that are flattened in row-major order to form a single new axis, following einops. A flattened axis is represented in the einx expression by wrapping the composed axes in parentheses. For instance, the output of the vectorized identity map

```
einx.id("a b c -> (a b) c", x)
```

is two-dimensional, and its first dimension corresponds to the original axes a and b flattened in row-major order (*i.e.*, a groups of b elements each).

We introduce a new type of axis composition that does not exist in einops, *i.e.* the *concatenated axis*, as multiple axes of multiple tensors concatenated along a single new axis. This allows representing many operations from Numpy-like notation (*e.g.*, np.{stack|concatenate|unstack|split}) as vectorized identity maps. A concatenated axis is represented in einx using the plus operator (+) with parentheses. For instance, the output of

```
einx.id("a b, a c -> a (b + c)", x, y)
```

is two-dimensional, and represents the concatenation of the input tensors along the second axis.

**Axis constraints**   Additional axis sizes may be passed as keyword arguments to einx functions, *e.g.*, if the input shapes of tensors do not fully constrain the lengths of all axes:

```
einx.id("(a b) c -> a b c", x, a=4)
```

**Anonymous axes**   For convenience, numerical axes may be used to specify the value of axes inline, and are equivalent to writing a new, unique axis name with a corresponding constraint:

```
einx.id("a b -> a b 3", x)
einx.id("a b -> a b c", x, c=3)
```

**Ellipsis**   We introduce a novel, generalized type of ellipsis ... that is placed immediately after an axis to indicate that it is expanded a variable number of times. The number of expansions is determined from the dimensionality of the input tensors and additional constraints. The following example illustrates the expansion of ellipses:

```
einx.add("b... i, b... j -> b... i j", x, y) # expands to ...
einx.add("b0 b1 i, b0 b1 j -> b0 b1 i j", x, y) # ... for 3D inputs
```

Ellipses also apply to composed axes. The following example expands a flattened axis in order to partition an $n$-dimensional tensor into a list of $n$-dimensional tiles with side-length ds:

```
einx.id("(s ds)... -> (s...) ds...", x, ds=4)
```

einx further allows writing an *anonymous* ellipsis without a preceding axis. In this case, einx introduces a new axis name in front of it.

Ellipses in einx are analogous to their role in languages such as Java, C++ and Swift: An ellipsis is placed after a parameter to indicate that the function or template accepts a variable number of arguments of that type. The actual number is determined from how many arguments are provided at a given call site. In contrast, anonymous ellipses are analogous to their usage in einsum and einops.

**Implicit output**   If possible, operations allow omitting the output and inferring it from the inputs instead, resulting in a more concise expression:

```
einx.sum("a [b]", x) # -> a          einx.add("a b c, c", x, y) # -> a b c
```

## 4.2   CHARACTERISTICS

**Universal**   einx decouples operations from their vectorization and applies consistent rules to express the vectorization independent of the specific operation. *Any* tensor operation may be vectorized with einx notation, and *any* vectorization representable in loop notation may also be expressed with einx notation. This makes einx a universal notation for tensor operations.

In practice, the universality allows invoking arbitrary operations, including those not part of einx's API. For instance, the following code creates an einx operation that vectorizes a custom Python function by internally using the vmap transformation from PyTorch (*c.f.* Sec. 2.3):

```
def myop(x, y): # Define a custom function
    return 2 * x + torch.sum(y)
einmyop = einx.torch.adapt_with_vmap(myop) # Convert to einx operation
```

Invoking the einx operation with

```
z = einmyop("a [c], b [c] -> a b [c]", x, y)
```

results in the same output as calling myop in loop notation:

```
for a in range(...): for b in range(...):
    z[a, b, :] = myop(x[a, :], y[b, :])
```

**Declarative**   Numpy-like notation follows an imperative programming model: It requires the programmer to express *how* to achieve the desired result, *e.g.*, involving reshaping, broadcasting, and transposing dimensions. In contrast, einx adopts a declarative approach similar to einsum, where the user specifies *what* the inputs and outputs look like, and allows the system to determine the required transformations. This is illustrated by the following example:

```
einx.add("a d e, c b e -> a b c d e", x, y)                # declarative
x[:, None, None] + np.transpose(y, (1, 0, 2))[None, :, :, None] # imperative
```

The former is often easier to read and write, and explicitly documents what the inputs look like *before* applying the operation and the outputs look like *after* applying the operation; both of which are not immediately visible in Numpy-like notation.

**Interpretable**   The definition of einx notation by analogy with loop notation provides an explicit interpretation of any given operation: The representation in loop notation clearly illustrates what output the operation will yield, while allowing for an underlying backend implementation that follows a different, more optimized algorithm.

## 4.3   PRACTICAL ADVANTAGES

In the following, we demonstrate several practical advantages of using einx with example operations. Additional examples can be found in Appendix C.

**Changing the shapes**   We consider a simple indexing operation in einx and Numpy-like notation where elements in the argument x are retrieved at positions stored in the argument y:

```
einx.get_at("[x] a, b -> b a", x, y)        torch.index_select(x, 0, y)
```

We now change the input and output shapes of this operation. einx allows varying the vectorization term to reflect these changes and keeps the entry-point fixed. In contrast, changing the shapes in Numpy-like notation necessitates switching to a different entry-point with a different signature and vectorization rules, or is not representable using a single entry-point at all:

```
# 1. Introduce axis a in 2nd parameter y -> switch to torch.take_along_dim
einx.get_at("[x] a, b a -> b a", x, y)     torch.take_along_dim(x, y, dim=0)
# 2. Introduce axis c -> no single entry-point in torch
einx.get_at("[x] b, c b a -> c b a", x, y)
# 3. Replace 1D indexing with 2D indexing -> no single entry-point in torch
einx.get_at("[x y] b, c b a [2] -> c b a", x, y)
```

**Silent failures**   einsum represents multiple elementary operations in a single entry-point:

```
np.einsum("ab,bc->ac", x, y)     einx.dot("a [b], [b] c -> a c", x, y)
np.einsum("ab->a", x)            einx.sum("a [b] -> a", x)
np.einsum("a,b->ab", x, y)       einx.multiply("a, b -> a b", x, y)
np.einsum("ab->ba", x)           einx.id("a b -> b a", x)
```

This potentially results in silent failures if a typo in the expression of one operation matches the signature of another operation. einx catches such errors by checking for the signature of the respective entry-point:

```
einsum("ij,jk->ik", x, y)                # succeeds -> dot-product along j
# Now introduce a typo:
einsum("ij,ik->ik", x, y)                # fails silently -> sum-reduction along j
einx.dot("i [j], [i] k -> i k", x, y) # fails loudly -> inconsistent brackets
einx.dot("i j, i k -> i k", x, y)     # fails loudly -> not a dot-product
```

**Clarity**   In complex operations, the undifferentiated definition of axes in einsum obfuscates which axes are summed along. In contrast, brackets in einx make the distinction clearly visible:

```
einsum("b q k h, b k h c -> b q h c", x, y) # Which axes are summed along?
einx.dot("b q [k] h, b [k] h c -> b q h c", x, y) # Only k is summed along!
```

## 4.4 Implementation

We provide an implementation of einx that compiles einx operations to function calls in a given tensor framework, *e.g.*, using Numpy-like or vmap notation (*c.f*. Sec. 2.3). The compilation creates an isolated code snippet that is transformed to a function object using Python's exec, cached on the first invocation, and reused on subsequent calls with the same signature. This results in no overhead compared to calling the framework functions directly, other than for cache lookup and during initialization (*c.f*. Appendix G). If used with just-in-time compilation such as jax.jit, the einx footprint disappears entirely.

As an example, the operation

```
einx.sum("a ([b] c)", x, c=4)
```

compiles to the following code when invoked with a Jax tensor of shape (8, 24) and requesting Numpy-like or vmap notation:

```
# backend="jax.numpylike"          # backend="jax.vmap"
import jax.numpy as jnp            import jax.numpy as jnp
def op(a):                         import jax
    a = jnp.reshape(a, (8, 6, 4))  b = jax.vmap(jnp.sum, in_axes=1, out_axes=0)
    a = jnp.sum(a, axis=(1,))      b = jax.vmap(b, in_axes=0, out_axes=0)
    return a                       def op(a):
                                       a = jnp.reshape(a, (8, 6, 4))
                                       a = b(a)
                                       return a
```

The compilation to Numpy-like notation uses features such as the axis parameter to express the vectorization, while vmap notation relies on the vmap transformation. We provide a description of how einx expressions are compiled to Python code in Appendix D, more examples of compiled code in Appendix E, and examples of verbose exceptions that are raised for syntax, shape and semantic errors in Appendix F.

## 5 Comparison with einsum and einops

In the following, we compare einx notation with einsum and einops notation and illustrate the distinctions by implementing an example tensor operation. We provide comparisons with other types of ein*-notations in Appendix A.

### 5.1 General comparison

Both einsum and einops do not recognize the role of vectorization in tensor operations, and contain design choices that are in contradiction with this insight:

- There is no distinction between vectorized axes and argument sub-tensor axes.
- The analogy with loop notation is not recognized or incorporated into the notation.
- einops.repeat and einops.reduce are framed as symmetrical in terms of adding or removing axes[1], despite the former applying *vectorization* to add an axis, and the latter applying an *elementary operation* to remove an axis.
- The naming of functions is not related to the underlying elementary operations: einsum is not called dot, einops.rearrange and einops.repeat are not called identity.
- einops.rearrange and einops.repeat compute the same elementary operation (*i.e*. identity map), but follow different vectorization behavior across different entry-points.

Unlike einsum and einops which support only few operations (*c.f*. Tab. 2), einx allows expressing any tensor operation and any vectorization by analogy with loop notation. It includes many notational improvements, such as generalized ellipses, axis concatenations, implicit outputs, a cleaner API and separation of elementary operations into individual entry-points. Our implementation further compiles expressions to isolated Python code snippets that are inspectable by the user and allow for different types of backend notations, such as Numpy-like or vmap notation.

---

[1] *"we made an explicit choice to separate scenarios of "adding dimensions" (repeat), "removing dimensions" (reduce) and "keeping number of elements the same" (rearrange)"* (Rogozhnikov, 2022a)

## 5.2 CASE STUDY: MULTI-HEAD ATTENTION

We consider the multi-head attention (MHA) operation (Vaswani et al., 2017) and compare implementations using (1) einx and (2) einsum, einops and Numpy-like notation if necessary. The axes b, q, k, h and c denote the batch, query, key, head and channel dimensions.

```
def attn(q, k, v, heads=1):                                              einx
    A = einx.dot("b q (h [c]), b k (h [c]) -> b q k h", q, k, h=heads)
    A = einx.softmax("b q [k] h", A / jnp.sqrt(q.shape[-1] / heads))
    return einx.dot("b q [k] h, b [k] (h c) -> b q (h c)", A, v)
```

```
def attn(q, k, v, heads=1):                                    einsum/einops/
    q = einops.rearrange(q, "b q (h c) -> b q h c", h=heads)      Numpy-like
    k = einops.rearrange(k, "b k (h c) -> b k h c", h=heads)
    v = einops.rearrange(v, "b k (h c) -> b k h c", h=heads)
    A = jnp.einsum("bqhc,bkhc->bqkh", q, k) / jnp.sqrt(q.shape[-1])
    A = jax.nn.softmax(A, axis=-2)
    output = jnp.einsum("bqkh,bkhc->bqhc", A, v)
    return einops.rearrange(output, "b q h c -> b q (h c)")
```

We make the following observations: (1) einx requires just three lines of code. einops additionally calls `einops.rearrange` due to `einsum` not supporting axis compositions. (2) The softmax operation in einx self-documents axis names and indicates that it is applied along the axis k. einops does not support softmax and uses Numpy-like notation with a positional `axis` argument. (3) einx indicates with brackets that the dot-products are applied along the axes c and k. einsum relies on an implicit convention and obfuscates which of the enumerated axes are reduced. (4) einx explicitly names the elementary operations, *i.e.* dot and softmax, rather than using the less clear name einsum.

In the MHA operation, a mask is optionally applied to the attention matrix:

```
qs, ks = jnp.arange(q.shape[1]), jnp.arange(k.shape[1])                 einx
mask = einx.greater_equal("q, k -> q k", qs, ks)
A = einx.where("q k, b q k h,", mask, A, -jnp.inf)
```

```
qs, ks = jnp.arange(q.shape[1]), jnp.arange(k.shape[1])        einsum/einops/
mask = qs[:, np.newaxis] >= ks[np.newaxis, :]                     Numpy-like
A = jnp.where(mask[np.newaxis, :, :, np.newaxis], A, -jnp.inf)
```

The element-wise operations are not supported in einops and must rely on Numpy-like notation which obfuscates both the semantics of axes and how they are aligned w.r.t. each other. In contrast, einx self-documents axis names and follows a declarative, rather than imperative style.

Following the decomposition of complex tensor operations described in Sec. 3.2, we consider an alternative implementation that represents the batched MHA shown above as an elementary, single-query, single-head attention operation and its separate vectorization:

```
def attn(q, k, v): # Define attention as an elementary operation          einx
    A = einx.dot("[c], k [c] -> k", q, k)
    A = einx.softmax("[k]", A / jnp.sqrt(q.shape[-1]))
    return einx.dot("[k], [k] c -> c", A, v)
einattn = einx.jax.adapt_with_vmap(attn) # Adapt to einx notation
# Vectorize along batch, query, and flattened head dimensions:
output = einattn("b q (h [c]), b [k] (h [c]), b [k] (h [c]) -> b q (h [c])",
                                                    q, k, v, h=heads)
```

## 6 CONCLUSION

We introduce einx, a universal notation for tensor operations. It follows a consistent set of rules that apply to any given operation, offers interpretability by analogy with loop notation, reduces the large API of existing Numpy-like frameworks to a small set of elementary operations, and allows for a clean, readable and writable expression of operations in code. The notation offers not only a useful coding tool, but a better model for thinking tensor operations. We provide an open source software package that implements einx in Python for commonly used tensor frameworks.

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

APPENDIX

We provide the following additional content in the appendix.

**Sec. A:** Comparison with other types of ein*-notation: `einops.{pack|unpack}`, eindex, einmesh

**Sec. B:** Comparison with Numpy-like notation

**Sec. C:** Additional examples of einx operations

**Sec. D:** Description of einx compiler

**Sec. E:** Additional examples of code snippets compiled for einx operations

**Sec. F:** Examples of einx exceptions

**Sec. G** Benchmark of einx's overhead

**Sec. H:** Usages statistics of related libraries

## A  COMPARISON WITH OTHER EIN*-NOTATIONS

**einops.pack, einops.unpack**   Rogozhnikov (2022b) introduces a new ein*-notation to einops that is implemented in `einops.{pack|unpack}` and allows expressing some concatenation and splitting operations. The following call flattens all but the first two dimensions of the input tensors and concatenates them along the third dimension:

```
einops.pack([x, y], "a b *")
```

However, the notation differs from the original einops notation, and further diverges from the declarative style where all inputs and outputs are documented explicitly. Instead, multiple arguments are represented using a single expression, and multiple varying sequences of axes are represented using the new `*` operator.

In contrast, concatenation and splitting in einx are expressed as special cases of the *vectorized identity map* using the concatenated axis composition (*c.f.* Sec. 4.1), retain the explicit and self-documenting style, support more vectorization patterns than `einops.{pack|unpack}`, and trivially allow inverting the operation by swapping input and output expressions:

```
einx.id("a b1, a b2 -> a (b1 + b2)", x, y)      einops.pack([x, y], "a *")
einx.id("a b1, b2 a -> a (b1 + b2)", x, y)      # no single entry-point in einops
einx.id("a (b1 + b2) -> a b1, b2 a", z, b1=4)   # no single entry-point in einops
einx.id("a, b -> a b (1 + 1)", x, y)            # no single entry-point in einops
```

**eindex**   Rogozhnikov (2023) proposes a notation that allows expressing gather, scatter and arg-operations. For example:

```
EX.gather(x, idx, "b h w c, [h, w] b -> b c")
einx.get_at("b [h w] c, [2] b -> b c", x, idx) # same operation in einx
```

The sub-expression `[h, w]` in eindex denotes an axis with length 2 in the tensor whose values are used to index the axes h and w of the value tensor. It must always appear as the first axis of the expression, diverges from the declarative style of the original notation, and does not generalize to other tensor operations. In contrast, indexing in einx follows the same notation as other operations, retains a more declarative style and supports more vectorization patterns:

```
einx.get_at("[h] c, [1] b -> b c", x, y)  EX.gather(x, y, "h c, [h] b -> b c")
einx.get_at("[h] c, b [1] -> b c", x, y)  # no single entry-point in eindex
einx.get_at("[h] c, b    -> b c", x, y)   # no single entry-point in eindex
```

**einmesh**   Jensen (2025) introduces an ein*-notation for meshgrid operations:

```
xs, ys = einmesh.LinSpace(0, 1, 10), einmesh.LinSpace(-1, 1, 20)
x, y = einmesh.numpy.einmesh("x y", x=xs, y=ys)
xy = einmesh.numpy.einmesh("x y *", x=xs, y=ys)
```

Mesh-grid operations are compositions of broadcasting and concatenation with existing generator functions such as `np.linspace`. As such, they are special cases of the *vectorized identity map* and expressible using `einx.id`:

```
xs, ys = np.linspace(0, 1, 10), np.linspace(-1, 1, 20)
x, y = einx.id("x, y -> x y, x y", xs, ys)
xy = einx.id("x, y -> x y (1 + 1)", xs, ys)
```

While einmesh requires knowledge of the concept and meaning of mesh-grids, the einx expression clearly self-documents the behavior without requiring the introduction of new concepts and documentation.

## B    COMPARISON WITH NUMPY-LIKE NOTATION

We observe that much of the complexity in Numpy-like notation stems solely from the way in which vectorization is expressed and impacts how users read and write tensor programs:

- Users have to learn many diverging rules for expressing vectorization, *e.g.*:
    - Operations over multiple inputs often rely on implicit broadcasting rules[2].
    - Some operations use parameters such as `axis` or `dim` (*e.g.* reduction with `torch.sum`, or vector-to-vector mapping with `torch.softmax`).
    - Indexing operations use, *i.a.*, advanced indexing rules[3].
    - Many operations (*e.g.* `np.{dot|matmul}`) follow function-specific rule sets.
    - Complex operations often require separate shape manipulation to align inputs and outputs with their signature (*e.g.* using `np.{transpose|squeeze|newaxis}`).
    - The rules sometimes conflict across different frameworks (*e.g.* `{tf|torch}.gather`).
- Function names and arguments alone often do not reflect the vectorization behavior without reading their documentation or writing comments, *e.g.*:
    - Which of `torch.{take|gather|index_select}` do I use to perform indexing in a given use case?
    - Which of `np.{matmul|dot|tensordot|inner}` do I use in a given use case?
- Understanding how a given operation is vectorized often incurs mental load, *e.g.*:
    - Which axes of `x` and `y` in the following expression are vectorized jointly or separately? `x[:, np.newaxis, :, np.newaxis] + y[:, :, np.newaxis, :]`
    - Which input and output axes in the following operation correspond to each other? `np.transpose(x, (2, 1, 3, 0))`

Harris et al. (2020) use the term *vectorization* only when describing element-wise operations in Numpy. In contrast, we follow a generalized view of vectorization that covers all mechanisms described above and is independent of any specific operation. This allows einx notation, which represents the vectorization of operations, to be applicable to *any* tensor operation and follow a *single* set of rules across operations. The universal nature of einx notation simplifies the large and complex API of Numpy-like notation, and reduces many Numpy-like operations to few einx operations.

## C    ADDITIONAL EXAMPLES OF EINX OPERATIONS

**Changing the operations**    In Numpy-like notation, some functions (*e.g.* `np.kron`) are provided for particular vectorization cases of an elementary operation (*e.g.* scalar multiplication), but similar specializations are not available for other elementary operations (*e.g.* scalar addition). In contrast, einx allows using analogous vectorization patterns across different operations:

```
einx.multiply("a..., b... -> (a b)...",          x, y) # Same as np.kron
einx.add     ("a..., b... -> (a b)...",          x, y) # kron-like add
einx.less    ("a..., b... -> (a b)...",          x, y) # kron-like less
einx.id      ("a..., b... -> (a b)... (1 + 1)", x, y) # kron-like stack
```

**Concatenation**    einx fully supports broadcasting and transposing shapes in concatenation operations, *e.g.*, to append a vector along the channel dimension of a batch of images:

---

[2]`https://numpy.org/doc/stable/user/basics.broadcasting.html`

[3]`https://numpy.org/doc/stable/user/basics.indexing.html#advanced-indexing`

```
einx.id("b c1 h w, c2 -> b (c1 + c2) h w", img, vec)
```

The same operation in Numpy-like notation requires separate manipulation of the shapes:

```
np.concatenate([img, np.broadcast_to(vec[None, :, None, None], \
    (img.shape[0], vec.shape[0], img.shape[2], img.shape[3])], axis=1)
```

einx similarly supports creating mesh-grids, which rely on a specialized entry-point in Numpy-like notation (*i.e.* np.meshgrid) and are not supported by a single entry-point in einops:

```
einx.id("x, y -> (1 + 1) x y", xs, ys) # Stacked along first axis
einx.id("x, y -> x y, x y", xs, ys) # Returned as separate tensors
```

The positional indices of arguments in some operations indicate how the arguments are used in the operation. Since axis concatenations change the number of arguments and therefore their positional indices, we only support axis concatenations in einx.id.

**Expanding composed axes**    We show the depth-to-space transformation (Shi et al., 2016) in einx and einops notation:

```
einops.rearrange(x, "b h w (c dh dw) -> b (h dh) (w dw) c", dh=4, dw=4)
einx.id("b s... (c ds...) -> b (s ds)... c", ds=4)
```

The axes b and c denote the batch and channel dimensions, h and w denote the spatial axes before, and (h dh) and (w dw) after the transformation. The ellipses allow for a joint representation of the spatial axes, resulting in a more concise expression, indicating similar treatment of spatial axes, and generalizing the operation to $n$ spatial dimensions. The following example shows a similar expression of a spatial mean pooling operation:

```
einops.reduce(x, "(h dh) -> h",
              reduction="mean", dh=4) # 1D
einops.reduce(x, "(h dh) (w dw) -> h w",
              reduction="mean", dh=4, dw=4) # 2D
einops.reduce(x, "(h dh) (w dw) (d dd) -> h w d",
              reduction="mean", dh=4, dw=4, dd=4) # 3D
einops.reduce(x, "(h dh) (w dw) (d dd) (z dz) -> h w d z",
              reduction="mean", dh=4, dw=4, dd=4, dz=4) # 4D
einx.mean("(s [ds])...", x, ds=4) # ND
```

**Multiple ellipses**    In einsum, multiple ellipses always refer to the same set of axes. In contrast, ellipses in einx expand custom axes and thereby allow representing multiple sets of axes:

```
einsum("... a, ... a -> ...", x, y) # Same set of axes
einx.dot("x... [a], x... [a] -> x...", x, y) # Same set of axes
einx.dot("x... [a], y... [a] -> x... y...", x, y) # Multiple sets of axes
```

**Flattened axis in einx.dot**    einsum and einops do not support using flattened axes for tensor contractions. In contrast, all operations in einx support flattened axes, *e.g.*, to express grouped linear layers in neural nets

```
# Regular linear layer
einx.dot("...    [in], [in]    out -> ...    out ", x, weights)
# Grouped linear layer: Same weights per group
einx.dot("... (h [in]), [in]    out -> ... (h out)", x, weights, h=heads)
# Grouped linear layer: Different weights per group
einx.dot("... (h [in]), [in] h out -> ... (h out)", x, weights, h=heads)
```

or the multi-head attention operation (*c.f.* Sec. 5.2).

**Multiple elementary operations**    As described in Sec. 3.2, some operations from Numpy-like notation are decomposable into different elementary operations. For instance, the sum-reduction y = np.sum(x, axis=1) is represented in loop notation as follows:

```
y[i] = sum(x[i, :]) # "sum" maps a vector to a scalar -> 1 vectorized axis
y[i] += x[i, j] # "+=" adds a scalar to a scalar -> 2 vectorized axes
```

This maps to two corresponding expressions in einx notation:

```
y = einx.sum("i [j] -> i", x) # 1 vectorized axis
y = einx.sum("i j -> i", x)   # 2 vectorized axes
```

Where possible, we support both types of expressions in an operation and indicate so in the documentation. The representation of einx.{dot|sum} as vectorized scalar operations (*i.e.*, without brackets) allows for compatibility with the corresponding operations in einsum and einops notation.

## D    DESCRIPTION OF EINX COMPILER

Our implementation compiles einx operations to isolated code snippets in Python which invoke framework functions based on the requested type of notation (*c.f.* examples in Sec. E). The compilation is performed in the following three steps.

**1. Abstract syntax tree**    In the first step, the string expression of a given einx operation is transformed to one abstract syntax tree (AST) for each input and output tensor. Nodes in the AST correspond to different sub-expressions such as axis lists, axis compositions, named or unnamed axes and ellipses. The transformation is done in the following stages:

1. Parse the string to a simple AST and check for syntax errors such as invalid literals, axis names, parentheses or brackets.

2. Expand all ellipses in the AST. The compiler first determines the number of expansions for each ellipsis using a system of equations that represent the constraints resulting among others from the input shapes or identical axis names across multiple ellipses. Each ellipsis is then replaced by $n$ copies of its child node where $n$ is its expansion number. For each copy, an incrementing counter is appended to all included axis names. Invalid ellipsis placement, *e.g.*, indicated by not finding a unique solution to the system of equations, results in a rank (*i.e.* dimensionality) error.

3. Determine the length of all axes in the expression and annotate the AST with the axis lengths. To achieve this, the compiler solves a system of equations that represent the constraints resulting among others from the input shapes, additional parameters and relation between nodes and their children (*e.g.*, the length of a flattened axis is equal to the product of the lengths of its child nodes). Inconsistent axis constraints, *e.g.*, due to input shapes not matching a given einx expression, result in a dimension error.

The final ASTs fully specify the shapes of all input and output tensors in the operation.

**2. Computational graph**    In the second step, a computational graph is built for the operation using the requested framework, notation, elementary operation, and shape ASTs. Nodes in the graph represent values (*e.g.*, tensors or Python values), and edges with input and output nodes represent function calls or other Python statements (*e.g.*, tensor operations in a given tensor framework).

The graph is built by passing tracers (*i.e.*, objects representing graph nodes) through a Python function that represents the algorithm for computing the given operation. The initial inputs are constructed as tracers representing the input tensors with the given shape ASTs. Each statement (*e.g.*, function call) with a set of input tracers constructs a new edge in the graph, and returns a new set of output tracers. The final graph is defined by a set of input and output tracers as well as edges representing the function calls and statements that make up the requested algorithm.

The algorithms for different types of operations are hard-coded based on the API of the backend framework and requested type of notation. Groups of operations often share parts of the implementation: For instance, most operations start by invoking a `reshape` operation to unflatten axes in the input tensors (*i.e.*, to remove flattened axis compositions), and end by flattening axes of the output tensors as determined by the output AST (*i.e.* to reintroduce flattened axis compositions). Some groups of operations, such as all element-wise operations, have identical implementations up to the innermost backend function that is invoked (*e.g.*, `np.{add|subtract|multiply|logical_and}`).

Finally, the graph is optimized using a set of simple heuristics, such as removing `reshape` operations where the input and output shapes are identical, or `transpose` operations where the order of input and output axes is identical.

**3. Python code snippet**    In the last step, the graph is transformed into an isolated Python code snippet. The operations are topologically sorted and transformed to lines of code by traversing the graph from output to input nodes. Variables are created starting from the name a and incrementing alphabetically, with names being reused if possible.

The entire code snippet is executed using Python's `exec`, and the object corresponding to the constructed operation is retrieved from the environment using Python's `eval`.

## E  ADDITIONAL EXAMPLES OF COMPILED CODE

The code snippet that is compiled for a given einx operation can be inspected by passing `graph=True` to the respective operation. In the following, we provide additional examples of compiled code for einx operations using the Jax framework.

**Example 1:** Transposition.
```
>>> x = jnp.zeros((10, 5))
>>> einx.id("a b -> b a", x, graph=True)
import jax.numpy as jnp
def op(a):
    a = jnp.transpose(a, (1, 0))
    return a
```

**Example 2:** Reshape.
```
>>> x = jnp.zeros((10, 5))
>>> einx.id("(a b) c -> a (b c)", x, b=2, graph=True)
import jax.numpy as jnp
def op(a):
    a = jnp.reshape(a, (5, 10))
    return a
```

**Example 3:** No-op.
```
>>> x = jnp.zeros((10, 5))
>>> einx.id("a b -> a b", x, graph=True)
def op(a):
    return a
```

**Example 4a:** Element-wise addition that uses Numpy-like broadcasting.
```
>>> x = jnp.zeros((2, 5, 6))
>>> y = jnp.zeros((4, 3, 6))
>>> einx.add("a d e, c b e -> a b c d e", x, y, graph=True)
import jax.numpy as jnp
def op(a, b):
    a = jnp.reshape(a, (2, 1, 1, 5, 6))
    b = jnp.transpose(b, (1, 0, 2))
    b = jnp.reshape(b, (1, 3, 4, 1, 6))
    c = jnp.add(a, b)
    return c
```

**Example 4b:** Element-wise addition that uses `jax.vmap` to vectorize `jnp.add`.
```
>>> x = jnp.zeros((2, 5, 6))
>>> y = jnp.zeros((4, 3, 6))
>>> einx.add("a d e, c b e -> a b c d e", x, y, graph=True,
                                        backend="jax.vmap")
import jax.numpy as jnp
import jax
a = jax.vmap(jnp.add, in_axes=(0, None), out_axes=0)
a = jax.vmap(a, in_axes=(1, None), out_axes=1)
a = jax.vmap(a, in_axes=(None, 0), out_axes=1)
a = jax.vmap(a, in_axes=(None, 1), out_axes=1)
a = jax.vmap(a, in_axes=(2, 2), out_axes=4)
```

**Example 5a:** Matrix multiplication that forwards to `jnp.einsum`.
```
>>> x = jnp.zeros((2, 3))
>>> y = jnp.zeros((3, 4))
>>> einx.dot("a [b], [b] c -> a c", x, y, graph=True)
import jax.numpy as jnp
def op(a, b):
    c = jnp.einsum("ab,bc->ac", a, b)
    return c
```

**Example 5b:** Matrix multiplication that uses `jax.vmap` to vectorize `jnp.dot`.

```
>>> x = jnp.zeros((2, 3))
>>> y = jnp.zeros((3, 4))
>>> einx.dot("a [b], [b] c -> a c", x, y, graph=True,
                                        backend="jax.vmap")
import jax.numpy as jnp
import jax
a = jax.vmap(jnp.dot, in_axes=(None, 1), out_axes=0)
a = jax.vmap(a, in_axes=(0, None), out_axes=0)
```

**Example 6:** Retrieve pixel colors from a batch of images.

```
>>> x = jnp.zeros((2, 128, 128, 3)) # batch of images
>>> y = jnp.zeros((50, 2)) # set of 50 pixels
>>> einx.get_at("b [h w] c, p [2] -> b p c", x, y, graph=True,
                                        backend="jax.vmap")
import jax
def a(b, c):
    return b[c[0], c[1]]
a = jax.vmap(a, in_axes=(0, None), out_axes=0)
a = jax.vmap(a, in_axes=(None, 0), out_axes=1)
a = jax.vmap(a, in_axes=(3, None), out_axes=2)
```

# F  EXAMPLES OF EINX EXCEPTIONS

Our implementation of einx raises verbose exceptions for syntax, shape and semantic errors. In the following, we provide several examples.

**Example 1:** Syntax error

```
>>> x = np.zeros((10, 5))
>>> einx.id("a b -> (a b", x)
...
SyntaxError: Found an opening parenthesis that is not closed:
Expression: "a b -> (a b"
                       ^
```

**Example 2:** Syntax error

```
>>> x = np.zeros((10, 5))
>>> einx.id("(b)a -> a b", x)
...
SyntaxError: The expression '(b)a' is not valid. Are you maybe missing a
whitespace?
Expression: "(b)a -> a b"
             ^^^^
```

**Example 3:** Bracket error

```
>>> x = np.zeros((10,))
>>> einx.sum("a [b] c -> a b", x)
...
SyntaxError: There are multiple occurrences of axis b with inconsistent bracket
usage:
Expression: "a [b] c -> a b"
               ^^^        ^
An axis may only appear with brackets or without brackets, but not both.
```

**Example 4:** Axis size error

```
>>> x = np.zeros((10, 5))
>>> einx.id("(a b) c -> a b c", x)
...
AxisSizeError: Failed to uniquely determine the size of the axes a, b. Please
```

```
provide more constraints.
Expression: "(a b) c -> a b c"
             ^ ^       ^ ^

The expression, tensor shapes and contraints resulted in the following
equation(s):
    (a b) c = 10 5
The operation was called with the following arguments:
  - Positional argument #1: Tensor with shape (10, 5)
```

**Example 5:** Ellipsis error

```
>>> x = np.zeros((10, 5))
>>> einx.id("(a b)... -> a b...", x)
...
RankError: Found an invalid usage of ellipses and/or constraints for the
axis a:
Expression: "(a b)... -> a b..."
             ^           ^

Please check the following:
 - Each axis name may be used either with or without an ellipsis, but not both.
 - The rank of a constraint must be equal to or less than the number of
   ellipses around the corresponding axis.
The following equation(s) were determined for the expression:
    (a b)... = 10 5
The operation was called with the following arguments:
  - Positional argument #1: Tensor with shape (10, 5)
```

Table 3: Overhead in milliseconds of using einx for three example operations with Numpy.

| Operation | Compilation (ms) | Cache retrieval (ms) |
|---|---|---|
| `einx.id("a h w c -> a c h w", x)` | $6.8 \pm 1.3$ | $0.058 \pm 0.001$ |
| `einx.dot("a [b], [b] c -> a c", x, y)` | $9.3 \pm 2.4$ | $0.070 \pm 0.007$ |
| `einx.add("a b (c d) e, (d e) f g h"` `"-> a b c d e f g h", x, y, d=2)` | $23.5 \pm 3.1$ | $0.077 \pm 0.003$ |

Table 4: Usage statistics of libraries in the context of tensor notations. *: einsum is implemented in different tensor frameworks, not a single repository. **: torchdim was upstreamed into the larger functorch repository on Aug 1, 2024. ***: No reliable search term.

| | Github stars | Github forks | Github files | Conferences usage |
|---|---|---|---|---|
| **ein\*-notation** | | | | |
| einsum | * | * | 431000 | 35.27% |
| einops | 9200 | 381 | 164000 | 21.19% |
| **Named tensors** | | | | |
| penzai | 1800 | 66 | 456 | 0% |
| torchdim | **331 | **10 | 243 | 0.01% |
| Haliax | 202 | 19 | 1000 | 0.02% |
| xarray | 4000 | 1200 | 184000 | 0.46% |
| **Other pointful notations** | | | | |
| Dex | 1600 | 114 | *** | 0% |
| Tensor Comprehensions | 1800 | 213 | 132 | 0% |
| Ein | 17 | 0 | *** | 0% |

## G  BENCHMARK OF OVERHEAD

einx compiles operations to function calls in a given tensor framework. The result of the compilation is cached on the first invocation, and reused on subsequent invocations with the same signature. The overhead of using einx compared to calling the framework functions directly thus consists of (1) the compilation on the first function call and (2) cache retrieval on subsequent function calls. Table 3 shows the overhead that einx incurs for three example operations using the Numpy backend. In all cases, the cache retrieval after initialization adds less than 0.1ms of overhead. When used with just-in-time compilation such as `jax.jit`, the overhead disappears entirely.

## H  USAGES STATISTICS OF RELATED LIBRARIES

Table 4 provides an overview on the usage statistics of related libraries in the context of tensor notations. We gathered the number of stars and forks of the respective Github repositories in September 2025. We further used the Github search to find usages of the libraries, and note the number of files returned by the search. We lastly gathered all 11898 publicly accessible Github repositories linked by papers published in the conferences CVPR 2023, CVPR 2024, CVPR 2025, ICCV 2023, ECCV 2024, ICLR 2023, ICLR 2024, ICLR 2025, NeurIPS 2023 and NeurIPS 2024, and note the percentage of papers that use the respective libraries by matching a regex term to the source code.

We make the following observations:

- Of the listed libraries, only einsum and einops are used by a significant number of papers ($> 20\%$) in machine learning conferences. All others are used by less than $1\%$ of papers.

- einsum, einops and xarray are used in $100\times$ more files on Github than all other libraries.

- xarray is used in many files on Github, but less than $1\%$ in machine learning conferences.

- Named tensor libraries and custom pointful notations have found little adoption in machine learning conferences.

