# OpenReview forum: "It's All Just Vectorization: einx, a Universal Notation for Tensor Operations"
_ICLR.cc/2026/Conference — ICLR 2026 Oral_

### Official Review · Reviewer_juN8 · 2025-10-27

**Soundness:** 3
**Presentation:** 3
**Contribution:** 2
**Rating:** 4
**Confidence:** 4

**Summary:**

The paper introduces einx, a notation and library for expressing tensor operations as vectorizations of elementary functions. The authors define a consistent syntax based on loop notation and implement it in Python for NumPy, PyTorch, and JAX backends. The goal is to replace the heterogeneous and sometimes confusing APIs of current tensor frameworks with a small, uniform set of composable operations. The paper is clearly written and technically sound.

**Strengths:**

- The paper is well organized and easy to follow.
- The notation is clearly defined and internally consistent.
- The implementation is described in sufficient technical detail and appears complete.
- The examples and tables illustrate coverage across many tensor operations.
- The case study on multi-head attention demonstrates syntactic conciseness.

**Weaknesses:**

- The contribution is about notation and implementation, not about new ML algorithms or empirical insights.
- The main problem addressed (API inconsistency) is largely syntactic and not shown to have a measurable impact on ML research or practice.
- No experiments, user studies, or adoption data support the claim that einx improves model development or reduces errors.
- The approach generalizes existing ideas from einsum and einops rather than introducing a new paradigm.
- The motivation section frames the issue as readability and conceptual elegance, but provides no evidence that current tools are a significant bottleneck.
- The comparison to other libraries is descriptive only, without performance or usability analysis.

**Questions:**

- Are there examples where einx allows expressing a model or computation that cannot be easily written with existing tools?
- Can you provide any data, even informal, on user adoption or code simplification in real ML workflows?
- How does einx handle contraction path optimization? Can it automatically determine and apply efficient evaluation orders for chained tensor products (e.g., einx.dot("m [a], [a b], [b] -> m", M, A, v)), or does it always execute operations in the explicit order implied by the expression?
- einx accepts sparse tensors for simple elementwise operations (e.g., scalar multiplication) but fails for contractions, additions, or reshapes because the PyTorch backend invokes dense operations (einsum, reshape, add(sparse, dense)). Could the authors clarify whether sparse interoperability is within the design scope of einx? Are there plans to support dispatching contractions to sparse-aware kernels (e.g., torch.sparse.mm), or is einx currently intended primarily for dense tensors?
- How much runtime overhead does the einx compilation and caching mechanism add compared to equivalent direct calls in PyTorch or NumPy, and are there cases where it limits backend optimizations (e.g., for dynamic shapes or JIT execution)?

---

> ### Author Response · Authors · 2025-11-21
> **Response to reviewer juN8 (1)**
>
> **W1: "The contribution is about notation and implementation, not about new ML algorithms or empirical insights."**
>
> We agree with this observation, but disagree that it represents a weakness of our work. We are aware that our paper does not follow the "typical" paper structure with method and experimental evaluation. Still, einx fundamentally impacts how machine learning models are mentalized, how machine learning code is written, and addresses problems that most machine learning practitioners are familiar with (including us). The fact that einx is already widely adopted in the community (see [Adoption of einx](https://openreview.net/forum?id=QqvQ3iAdpC&noteId=brHSSrpTHY)) supports this judgement.
>
> Furthermore, ICLR does accept papers that provide theoretical insight rather than empirical insight. This includes the work introducing einops at ICLR 2022 which has a similar structure as ours: It introduces (1) a notation, and (2) a software implementation. The final acceptance decision of the review for the einops paper (https://openreview.net/forum?id=oapKSVM2bcj) explicitly acknowledges that "[t]his is a paper about design, not about models or algorithms", and that such a paper is still relevant since the job of ICLR is to "expose researchers and practitioners in machine learning to ideas and techniques that may advance their research and practice".
>
> **W2.1: "The main problem addressed (API inconsistency) is largely syntactic"**
>
> We kindly disagree that the main contribution of einx is largely syntactic. einx differs fundamentally from previous notations by explicitly recognizing the role that vectorization plays in tensor operations. This is a theoretical insight, and it is what allows einx notation to be universally applicable across tensor operations (see [einx notation is universal](https://openreview.net/forum?id=QqvQ3iAdpC&noteId=brHSSrpTHY)).
>
> Beyond the literal syntax of einx, it is also about the proposed mental model for operations that einx promotes. For instance, when designing layers in machine learning models, a paradigm inspired by Numpy-like notation might ask something like
>
> > "Do I want to use a np.dot operation here, or maybe a np.tensordot operation, or the inner product (np.inner), or a matrix multiplication (np.matmul), or maybe even a simple multiplication (np.multiply)?" or "Do I want to gather values from this tensor (torch.gather), or maybe perform index selection from this tensor (torch.index_select), or take some values along dimensions of this tensor (torch.take)?"
>
> while a paradigm inspired by einx notation and our observations on vectorization would always ask:
>
> > "What elementary operation do I want to perform here (e.g., einx.dot or einx.get_at?), and how do I want it to be vectorized along the tensor axes?"
>
> **W2.2: "not shown to have a measurable impact on ML research or practice" W3: "No experiments, user studies, or adoption data support the claim that einx improves model development or reduces errors." Q2: "Can you provide any data, even informal, on user adoption or code simplification in real ML workflows?" W5: "no evidence that current tools are a significant bottleneck"**
>
> einx has been publicly available for 2 years, and has already found wide-spread adoption in the machine learning community (see [Adoption of einx](https://openreview.net/forum?id=QqvQ3iAdpC&noteId=brHSSrpTHY)). While it is difficult to quantify or measure the impact directly, the existing adoption of einx does provide evidence of its usefulness and impact on machine learning research and practice, as well as a "bottleneck" of other, existing tools.
>
> We also note that einx by design is able to catch errors in tensor code that einsum and einops are not able to catch (see lines 742ff in the Appendix).

---

> ### Author Response · Authors · 2025-11-21
> **Response to reviewer juN8 (2)**
>
> **W4: "The approach generalizes existing ideas from einsum and einops rather than introducing a new paradigm"**
>
> We kindly disagree that einx is a generalization of einsum and einops. einx follows an entirely different paradigm that is based on our observations on vectorization in tensor operations. In particular, einsum and einops contain design choices that distinctly contradict these observations:
> - All axes in their expressions - both vectorized and sub-tensor argument axes - are in the same category and simply viewed as analogous to indices in mathematical notation. Yet, a distinction of axes is necessary to represent arbitrary cases of vectorization.
> - The two functions `einops.rearrange` and `einops.repeat` exist, despite them computing the same elementary operation (identity-map) and differing only in their vectorization.
> - The two functions `einops.repeat` and `einops.reduce` are framed as symmetrical, despite the former applying vectorization to add an axis, and the latter applying some elementary operation to remove an axis.
> - The naming of functions is not related to the underlying elementary operations, and does not indicate the decomposition of tensor operations into elementary operations and their vectorization: `einsum` is not called `dot`, `rearrange` and `repeat` are not called `id`/`identity`. The functions instead follow the unrelated principle: "[W]e made an explicit choice to separate scenarios of “adding dimensions” (`repeat`), “removing dimensions” (`reduce`) and “keeping number of elements the same” (`rearrange`)" (Rogozhnikov et al. 2022a).
>
> **W6: "The comparison to other libraries is descriptive only, without performance or usability analysis."**
>
> The existing adoption of einx in the community (see [Adoption of einx](https://openreview.net/forum?id=QqvQ3iAdpC&noteId=brHSSrpTHY)) provides some evidence for its preference over other libraries. While the focus of our paper is on the notation and paradigm, rather than our software implementation, we have added a section in the appendix detailing the performance overhead for compilation and cache lookup (see Q5.1). We further provide a detailed comparison with several existing types of ein*-notations (einops.pack, eindex, einmesh) in Appendix A, including example operations expressed in both notations for comparison.
>
> **Q1: "Are there examples where einx allows expressing a model or computation that cannot be easily written with existing tools?"**
>
> The example in line 367 cannot be expressed easily with existing types of notation. There are many more examples in Appendix A and C.
>
> **Q3: "How does einx handle contraction path optimization?"**
>
> einx compiles expressions to function calls in a given backend framework. For instance, in the Jax backend, tensor contractions (i.e. einx.dot) internally call either a jnp.dot with jax.vmap, or jnp.einsum. Therefore the contraction paths in einx.dot are determined internally by the Jax framework, not by einx. Contraction paths are also only relevant in tensor contractions, not other operations. Sec. 4.3 and Appendix E show many examples of compiled code for einx operations. For example, the operation
> ```
> einx.dot("a [b], [b] c -> a c", x, y, backend="jax.vmap")
> ```
> compiles to the following code:
> ```
> import jax.numpy as jnp
> import jax
> a = jax.vmap(jnp.dot, in_axes=(None, 1), out_axes=0)
> a = jax.vmap(a, in_axes=(0, None), out_axes=0)
> ```
> **Q4: "Could the authors clarify whether sparse interoperability is within the design scope of einx?"**
>
> Yes, sparse tensors are supported by the notation itself, although support is currently not implemented by our software library. einx determines the backend to use for an operation from the tensor types it is given (and an additional backend argument if given), so a future version of einx could invoke sparse backend functions for sparse tensors, similar to how it currently chooses a Jax backend for Jax tensors and the Numpy backend for Numpy tensors.

---

> ### Author Response · Authors · 2025-11-21
> **Response to reviewer juN8 (3)**
>
> **Q5.1: "How much runtime overhead does the einx compilation and caching mechanism add"**
>
> While the focus of our paper is on the notation and paradigm, rather than our software implementation, we have added a section in the appendix detailing the performance overhead for compilation and cache lookup. The overhead at initialization (i.e. the first time an operation is called), and for the cache-read (i.e. any subsequent time the operation is called with the same signature) are benchmarked for three expressions:
> ```
> Benchmark for einx.id("a h w c -> a c h w", x):
> init=6.7877ms ± 1.3156ms
> cacheread=0.0582ms ± 0.0013ms
>
> Benchmark for einx.dot("a [b], [b] c -> a c", x, y):
> init=9.2989ms ± 2.3824ms
> cacheread=0.0699ms ± 0.0067ms
>
> Benchmark for einx.add("a b (c d) e, (d e) f g h -> a b c d e f g h", x, y, d=2):
> init=23.4886ms ± 3.0594ms
> cacheread=0.0772ms ± 0.0026ms
> ```
> After initialization, einx adds an overhead of <0.1ms for each operation. If used with just-in-time compilation such as jax.jit, the overhead disappears entirely, since Jax itself traces through the function calls made by einx and therefore invokes the einx operations only a constant, small number of times.
>
> **Q5.2: "are there cases where it limits backend optimizations (e.g., for dynamic shapes or JIT execution)"**
>
> einx compiles expressions to function calls in a given backend, so there are no limitations beyond those that arise from going through the cache retrieval and calling these framework functions directly. JIT compilation such as with jax.jit is seamlessly compatible with einx, since it traces through the backend function calls made by einx internally, such that the einx footprint disappears entirely. Dynamic shapes are supported by the notation itself, but not currently by our implementation of it. This might be added in a future version.

---

> ### Comment · Reviewer_juN8 · 2025-11-25
>
> Thank you for the detailed rebuttal and for addressing each of my questions. However, my core concerns remain largely unresolved.
>
> Novelty and conceptual contribution
> The idea that many tensor operations differ only in how they apply a common lower-dimensional computation across axes is correct, but this observation is already central to several existing tensor abstractions, including JAX vmap and batching rules, XLA and MLIR tensor dialects, Tensor Comprehensions, functional tensor languages, and polyhedral compilation frameworks. These systems explicitly treat vectorization as a general lifting mechanism over primitive operators. The rebuttal does not show that the claim "it is all just vectorization" is new relative to this prior work or formalize the idea in a way that advances theory. As a result, the contribution still reads mainly as a notation and design proposal, rather than a scientific contribution of the kind typically expected at ICLR.
>
> Lack of empirical evidence
> The paper repeatedly suggests that einx improves readability, reduces complexity, and helps avoid shape errors, but it provides no empirical or usability evidence to support these effects. Adoption numbers are useful but cannot substitute for evaluation, and they do not show that einx improves correctness or simplifies model development in practice. The rebuttal refers to the precedent of einops, but that work focused on a much narrower problem of tensor rearrangement and axis manipulation, which made the lack of experiments more acceptable at the time. By contrast, einx proposes a broader and more ambitious "universal" programming abstraction, so stronger evidence is needed. My expectations for ICLR 2026 are higher, and the absence of empirical validation remains a key gap.
>
> Universality and scope
> The rebuttal argues that the notation is universal because any operation can be expressed as a lower dimensional computation lifted over extra axes. This vectorization view works for many dense, fixed-shape common tensor operations, but it does not cover all cases. Operations involving sparse or layout dependent tensors, contraction path optimization, dynamic shape dependent behavior, and other backend specific rules do not reduce cleanly to simple axis wise lifting. Since einx ultimately relies on backend semantics for these cases, the practical coverage is narrower than the universal framing suggests, and some of the stronger claims feel ahead of what is currently supported.
>
> Overall, I appreciate the clarifications, and I agree that einx is likely valuable for practitioners, as suggested by the informal adoption statistics. However, the fundamental concerns around novelty, evaluation, and scientific contribution remain. My assessment therefore stays the same. Still, given the additional informal adoption data you provided, I am willing to raise the score to a 6.

---

> ### Author Response · Authors · 2025-12-03
> **2nd Response to reviewer juN8 (1)**
>
> Thank you for your feedback and response. We'd like to address some remaining disagreement on our side regarding the statements w.r.t. novelty and universal applicability of einx.
>
> > **The idea that many tensor operations differ only in how they apply a common lower-dimensional computation across axes [...] is already central to several existing tensor abstractions. [...] XLA [...] Tensor Comprehensions [...] These systems explicitly treat vectorization as a general lifting mechanism over primitive operators**
>
>
> - **XLA (StableHLO)** does not treat vectorization as a general *(i.e. universal, consistent)* lifting mechanism: Different operations still follow different interfaces and rules to express their inherent vectorization, and there is no recognition that the underlying commonality between these functions is actually just their vectorization.
>
>     **What XLA does:** It provides generalized interfaces for some groups of Numpy-like functions, e.g. `dot_general` for `np.dot`, `np.tensordot`, `np.matmul`, and `np.inner`, and `gather` for indexing functions.
>
>     **What XLA doesn't do:** It does not provide a universal perspective on vectorization *across different elementary operations*. For instance, `dot_general` represents its vectorization using the parameters `lhs_batching_dimensions`, `rhs_batching_dimensions`, `lhs_contracting_dimensions`, `rhs_contracting_dimensions`, while `gather` follows an entirely different interface consisting among others of parameters `offset_dims`, `collapsed_slice_dims`, `operand_batching_dims`, `start_indices_batching_dims` that follow different rules than the vectorization in `dot_general` (see https://github.com/openxla/stablehlo/blob/main/docs/spec.md). The fact these different interfaces essentially are just different ways to express vectorization is neither recognized, nor incorporated into the design. In contrast, einx uses a single notation to express the vectorization across *any* elementary operation (including `einx.dot` and `einx.get_at`); i.e., it follows a universal perspective on vectorization.
>
> - **Tensor Comprehensions (TC)** does not treat vectorization as a general lifting mechanism *over arbitrary operations*. Instead, it is inherently restricted to scalar operations as elementary operations.
>
>     TC notation "borrows from the Einstein notation (a.k.a. summation convention" (Vasilache et al., 2018), and relies on the same implicit convention as einsum/einops that "indices that appear on the [input] of an expression but not on the [output] are assumed to be reduction dimensions". Similar to einsum/einops, TC notation therefore does not distinguish between vectorized axes and argument sub-tensor axes: All axes in TC notation are vectorized axes. Consequently, elementary operations that are vectorized in TC notation *may only operate on scalars*, but not higher-dimensional tensors (e.g., as in many linear algebra functions such as np.linalg.solve, softmax or other neural net layers).
>
>     For example, the following matrix-vector product in TC notation
>     ```
>     def mv(float(M,K) A, float(K) x) → (C) {
>         C(i) = 0
>         C(i) += A(i,k) * x(k)
>     }
>     ```
>     vectorizes the scalar multiply-then-add operation (`*` and `+=`), but there is no support to similarly vectorize a non-scalar operation. In contrast, einx identifies argument sub-tensor axes using brackets (analogous to `:` in loop notation) and thereby allows vectorizing arbitrary (scalar and non-scalar) operations.

---

> ### Author Response · Authors · 2025-12-03
> **2nd Response to reviewer juN8 (2)**
>
> > **Universality [...] This vectorization view works for many dense, fixed-shape common tensor operations, but it does not cover all cases. Operations involving sparse or layout dependent tensors, contraction path optimization, dynamic shape dependent behavior, and other backend specific rules do not reduce cleanly to simple axis wise lifting**
>
> We kindly disagree and want to emphasize that both our view on vectorization and our einx notation are independent of (1) the underlying layout of tensors in memory (e.g., dense or sparse) and (2) algorithms that execute tensor operations on hardware and their implicit optimization.
>
> **Regarding (1) tensor layout:** A tensor in einx is simply viewed conceptually as a mapping from coordinate tuple to (float-, int-, etc) value, regardless of whether it is represented in memory using a dense layout (where each value is stored explicitly), or using a sparse layout (where most coordinates map to a zero value and only few values are stored explicitly). einx notation is not in any sense "more" applicable to dense layout tensors than to sparse layout tensors. einx notation further is not more applicable to fixed-shape tensors (whose dimension sizes are known at compile time) than to tensors with dynamic shapes (whose dimension sizes are not known at compile time).
>
> Our attached software implementation of einx aims to support use cases with most major tensor frameworks and backends. This currently does not include sparse tensors (insofar as they are not already supported by entry-points that also cover dense operations) and dynamic tensor shapes (which are also not supported, e.g., by Numpy and Jax). This does not however touch on the notation itself which is independent of the tensor layout and dynamic or static shapes. In the future, we might add support to our einx software for sparse or dynamic-shape tensors.
>
> **Regarding (2) operation's implementation**: A vectorized tensor operation is defined in einx by analogy with loop notation - the output is identical to the result of the loop notation, but there are no restrictions on how the operation is implemented using the underlying backend. Such implementation details are typically not in the scope of tensor notations - e.g. Numpy's `np.dot` will defer to BLAS routines if possible, but this detail is hidden to the end-user. Similarly, the contraction path of tensor contractions does not impact the output of the operation (other than due to numerical issues), but rather is an implementation detail that is determined implicitly to optimize computation time, memory usage, or other performance metrics.
>
> As argued similarly in Sec. 4.2, tensor notations ideally are *declarative* (i.e. stating *what* should be computed) rather than *imperative* (i.e. stating *how* something should be computed), and contraction path optimization concerns the *how* rather than the *what*.

---

### Official Review · Reviewer_bcTx · 2025-10-29

**Soundness:** 4
**Presentation:** 3
**Contribution:** 4
**Rating:** 8
**Confidence:** 4

**Summary:**

This paper proposed a new extended grammar for describing tensor operations that generalizes those common in `einsum` and `einops`. The authors cast vectorization as a general transformation that lifts lower-order operations to higher-order operations with a pointful (instead of pointless) style that is more declarative. It can naturally subsume all kinds of indexing (gather, scatter, index_select); tensor contraction (multiply, inner, outer, kronecker product, etc.); and all kinds of reshaping operations (broadcast, repeat, squeeze, etc.). Additionally, the authors also included a `(a+b)` notation that covers stack and concat.

The case study with multi-head attention is pretty illuminating. This library would be beneficial to all machine learning and scientific computation practitioners.

**Strengths:**

- A good generalization of `einsum` and `einops` that would be beneficial to all ML practitioners.
- Paper is clearly written, with lots of examples to showcase the semantics the proposed grammar.

**Weaknesses:**

- The semantics of `[x]` is not clear enough: at sometimes it is for axes to be contracted; sometimes it can be used in a `get_at` expression whose semantics is a bit vague.

**Questions:**

- In the abstract, please cite `einsum` and `einops`.
- L165 Named tensors: Missing citations to include:
  - T Chen (2017): Typesafe abstractions for tensor operations. https://dl.acm.org/doi/10.1145/3136000.3136001
  - D Chiang, A M Rush, B Borak (2021): Named Tensor Notation. https://arxiv.org/abs/2102.13196
- L309: The semantics of `[ ]` is vague: it seems to indicate matched axes for tensor contraction, but sometimes it has other uses: please clarify

---

> ### Author Response · Authors · 2025-11-21
> **Response to reviewer bcTx**
>
> **W1/Q3: "semantics of brackets"**
>
> Axes without brackets are vectorized, and axes with brackets are sub-tensor argument axes and forwarded to the elementary operation. This distinction can be illustrated by comparing the einx notation and loop notation representation of the same operation:
> ```
> # einx notation
> z = einx.SOME_OPERATION("a [x y], b [i] -> a b", tensor1, tensor2)
>
> # loop notation
> for a in range(...): for b in range(...):
>     z[a, b] = SOME_OPERATION(tensor1[a, :, :], tensor2[b, :])
> ```
> The axes marked with brackets in einx notation are simply "kept" in the loop notation and are not indexed (indicated by the `:`). Consequently, the are forwarded to the elementary operation. In this case, the elementary operation is given a tensor with shape (x, y), and another tensor with shape (i), and it returns a scalar. Brackets are necessary to distinguish which axes are indexed in loop notation (i.e. vectorized axes), and which are forwarded to the elementary operation (i.e. sub-tensor argument axes).
>
> Crucially, this is regardless of what computation the operation performs. Brackets do not have a different meaning when expressing tensor contractions and when expressing indexing operations (e.g. get_at). This is due to the universal nature of the notation.
>
> An example of a tensor contraction with brackets:
> ```
> einx.dot("a [b], [b] c -> a c", t1, t2)
> ```
> The elementary operation is given two vectors with shape (b) and returns a scalar.
>
> An example of an indexing operation with brackets:
> ```
> einx.get_at("[h w] channels, points [i] -> points channels", image, points)
> ```
> The elementary operation is given a 2D image with shape (h, w) and a single coordinate vector with shape (i) (i is equal to 2 due to indexing a 2D image) and it returns a single scalar.
>
> **Q1: "In the abstract, please cite einsum and einops."**
>
> To our best knowledge, citations in the abstract are discouraged, since the abstract must be able to stand on its own. We checked several papers from ICLR 2024 at random and did not find any with citations in the abstract. We cite einsum and einops several times throughout our paper, starting in the introduction.
>
> **Q2: "Missing citations"**
>
> "T Chen: Typesafe abstractions for tensor operations": We have added the reference in line 165.
>
> "D Chiang, A M Rush, B Borak: Named Tensor Notation" is already in our references (see line 509). We cite the TMLR version from 2023 rather than the arxiv version from 2021.

---

### Official Review · Reviewer_GyJP · 2025-11-01

**Soundness:** 2
**Presentation:** 3
**Contribution:** 3
**Rating:** 6
**Confidence:** 4

**Summary:**

The paper proposes einx, a “universal” notation for tensor operations that extends einsum. It also claims other features, including declarative and Interpretable. The paper compares einx with einops using multi-head attention as a case study.

**Strengths:**

+ Brings many NumPy-like operations under a unified function signature.
+ The declarative nature of the notation is advantageous; for instance, Line 374 demonstrates how the necessary permutation is inferred and performed implicitly.

**Weaknesses:**

- Section 3 discusses tensor ops and vectorization, but lacks formal definitions of lower-order vs higher-order vectorization; this makes the argument hard to follow
- The core conclusion in Section 3 —“It’s all just vectorization”—reflects a well-known aspect of tensor computation (same computation applied across slices). The paper does not clearly present how this finding leads to the specific design choices in the einx signature.
- It’s unclear, in notation and capability, how the proposed {vectorization}/bracket syntax differs in practice from PyTorch einsum (or its common extensions).
- Table 1 categorizes Numpy-like notations into 4 groups. It might be interesting to discuss the intuition of such a classification.

**Questions:**

1. How does the finding about vectorization lead to the design of einx?
2. What problems do brackets solve that einsum cannot, and how often do those cases arise in practice?
For example, this notation seems unnecessary in many cases in the paper.
Line 358, einx.sum("a [b]", x) seems equivalent to torch.einsum(“a b -> a”, x).
Line 308, einx.dot("a [b], [b] c -> a c", x, y)  seems equivalent to torch.einsum("a b, b c", x, y)
Line 472, einx.softmax("[k]", A) seems equivalent to torch.softmax(a, axis=1).
3. What are the key differences between einx and PyTorch einsum (or other einsum extensions) in terms of notation design? Which operations can einx express that others cannot, beyond the softmax example above?
4. What is the scope? Is it really universal?
a. Convolution operators are important; why are they not covered or discussed in this paper? There are works extending einsum to convolution operations [1][2].
b. How does einx represent pixel shuffle?
c. How does einx represent jacobi-2D?
5. Table 1 classifies Numpy-like operations into four categories. What is the intuition for such classification?
6. How can the proposed classification (Table 1) be extended to cover custom operations? Can it serve as a framework or guideline for defining new operations within einx?
7. Section 5.2 presents multi-head attention as an example; however, similar high-level fused attention layers already exist. Please include additional examples where einx clearly shortens code, improves readability, or demonstrates other unique advantages.

[1] Dangel, Felix. "Convolutions and More as Einsum: A Tensor Network Perspective with Advances for Second-Order Methods." Advances in Neural Information Processing Systems 37 (2024): 96671-96727.

[2] Rabbani, Tahseen, et al. "conv_einsum: A Framework for Representation and Fast Evaluation of Multilinear Operations in Convolutional Tensorial Neural Networks." arXiv preprint arXiv:2401.03384 (2024).

---

> ### Author Response · Authors · 2025-11-21
> **Response to reviewer GyJP (1)**
>
> **W1: "formal definitions of lower-order vs higher-order vectorization"**
>
> We use the terms "higher-order operations" and "lower-order operations" merely to refer to tensor operations on higher-dimensional and lower-dimensional tensors, respectively. We have adapted the text to make this more clear.
>
> For example, the title
>
> *3.1 Vectorizing lower-order operations to higher-order operations*
>
> refers to taking an operation which processes some tensors, and vectorizing it to yield a new operation which processes higher-dimensional tensors than the original.
>
> **W2.1: "The core conclusion in Section 3 —“It’s all just vectorization”—reflects a well-known aspect of tensor computation"**
>
> We kindly disagree that our core conlusion of Sec. 3 "It's all just vectorization" is well-known. Numpy-like and ein*-notations - the most widely used notations for tensor programming in machine learning - neither recognize, nor incorporate this observation in their design (see [Novelty](https://openreview.net/forum?id=QqvQ3iAdpC&noteId=brHSSrpTHY)). For example, all of np.dot, np.tensordot, np.matmul, and np.inner perform the same elementary dot-product, and their difference lies solely in how they are vectorized (c.f. Tab. 1 and Sec. 3.2). Yet, these operations are presented as different tensor operations without any recognition of their commonality. The fact that the difference between these operations is just their vectorization ("It's all just vectorization") is neither recognized, nor incorporated into the notation.
>
> **W2.2/Q1: "How does the finding about vectorization lead to the design of einx?"**
>
> The idea of einx is to represent the vectorization of operations entirely using einx notation. This has several advantages:
> - Have a single set of notational rules to express vectorization, instead of many inconsistent, operation-specific rules.
> - Have just one entry-point for each elementary operation.
> - Have a notation that applies to new, custom operations without the need to define new operation-specific vectorization rules for them.
>
> **W3/Q3.1 "unclear [...] how the proposed [...] syntax differs in practice from PyTorch einsum (or its common extensions)."**
>
> - einx notation is universal, i.e. it supports any operation, while einsum and its common extensions are not. See Q3.2 for examples.
> - There are many further practical details that make einx notation superior to einsum/einops, including implicit outputs, and error handling - see Appendix C for more details.
>
> **W4: "Table 1 categorizes Numpy-like notations into 4 groups. It might be interesting to discuss the intuition of such a classification." Q5: "Table 1 classifies Numpy-like operations into four categories. What is the intuition for such classification?"**
>
> The grouping of multiple functions in the left column of Tab. 1 stems merely from them computing the same elementary operation when factoring out their vectorization (c.f. Sec. 3.2). This is indicated by mapping to the same einx function in the right column:
> - einx provides just one entry-point per elementary operation.
> - The vectorization is expressed entirely using the vectorization string in einx.
> - The difference between functions per group in the left column reduces to differences in the vectorization string in the right column.
>
> The difference between many functions in Numpy-like notation (and therefore much of the complexity of Numpy-like notation) is solely due to their different vectorization - it's all just vectorization. The fact that they are indeed just different vectorized variants of the same elementary operation is, however, not recognized in, nor incorporated by Numpy-like notation.
>
> **Q6: How can the proposed classification (Table 1) be extended to cover custom operations?**
>
> This depends on the nature of the custom function that is under consideration. If it is merely a vectorized variant of an existing elementary operation, it would fall into its respective group. If it is not, it would correspond to a new group.

---

> ### Author Response · Authors · 2025-11-21
> **Response to reviewer GyJP (2)**
>
> **Q2: "What problems do brackets solve that einsum cannot"**
>
> Brackets (i.e. a way in which vectorized and sub-tensor argument axes are distinguished) are a prerequisite for representing arbitrary types of vectorization at all. For instance, to define the operation
> ```
> z = einx.SOME_OPERATION("a [b] -> a", x)
> ```
> analogous to the loop notation
> ```
> for a in range(x.shape[0]):
>     z[a] = SOME_OPERATION(x[a, :])
> ```
> we have to know which axes will be looped over (i.e. vectorized axes), and which will be forwarded to the elementary operation (i.e. sub-tensor argument axes, indicated with `:` in loop notation, and brackets in einx notation).
>
> Brackets thus allow representing arbitrary kinds of vectorization, while einsum cannot represent either arbitrary types of vectorization, or arbitrary operations.
>
> Brackets further have practical advantages such as visibly indicating which axis an operation is applied along (c.f. lines 755ff) and allowing for better error handling (c.f. lines 742ff).
>
> **Q3.2: "Which operations can einx express that others cannot, beyond the softmax example above?"**
>
> *Any* operation can be invoked using einx syntax, even custom ones. Examples of operations supported by einx but not by einsum and its common extensions are:
> - Elementwise operations such as addition (einx.add) or masking (einx.where)
> - Many vectorization cases of indexing operations (einx.get_at): See lines 637-638
> - Many vectorization cases of concatenation and splitting operations (einx.id): See lines 624-626
> - Arbitrary, new functions: See [einx notation is universal](https://openreview.net/forum?id=QqvQ3iAdpC&noteId=brHSSrpTHY)
>
> **Q4.1: "What is the scope? Is it really universal?"**
>
> Yes, einx is universal because it can be used to express the vectorization of *any* tensor operation, and because it allows for arbitrary tensor shapes and vectorization cases by analogy with loop notation. See [einx notation is universal](https://openreview.net/forum?id=QqvQ3iAdpC&noteId=brHSSrpTHY) for a more detailed discussion.
>
> **Q4.2a: "conv operator"**
>
> Convolutions are supported by einx notation. For instance, a depthwise convolution vectorizes along the batch and channel axes of an image tensor and is expressed as follows:
> ```
> # same kernel for all channels:
> einconv("b [image_height_in image_width_in] c, [kernel_height kernel_width]   -> b [image_height_out image_width_out] c", image, kernel)
>
> # different kernels for different channels:
> einconv("b [image_height_in image_width_in] c, [kernel_height kernel_width] c -> b [image_height_out image_width_out] c", image, kernel)
> ```
>
> einx does not have a dedicated convolution function, but convolutions are still representable using adapters such as `einx.jax.adapt_with_vmap`.
>
> **Q4.2b: "How does einx represent pixel shuffle?"**
>
> We show the pixel shuffle operation with einx in Appendix C at line 724 (we use the name "depth-to-space transformation").
>
> **Q4.2c: "How does einx represent jacobi-2D?"**
>
> We are not aware of a "jacobi-2D" operation. Could you provide a reference to a function in Numpy or another tensor framework?
>
> **Q7.1 "Section 5.2 presents multi-head attention as an example; however, similar high-level fused attention layers already exist"**
>
> There are indeed high-level fused layers in tensor frameworks such as `torch.nn.functional.scaled_dot_product_attention` to represent the MHA operation. However, these were implemented *after* the introduction of the MHA operation - the operation was thus not expressible in a similarly concise manner before that. In contrast, einx's universal nature allows for the concise expression of MHA even without being specifically designed for it. Analogously, einx can be used to express complex operations for which dedicated functions in `torch.nn.functional.*` have not yet been implemented.
>
> **Q7.2: "Please include additional examples where einx clearly shortens code, improves readability, or demonstrates other unique advantages."**
>
> Appendix C provides many examples where einx notation is superior to existing notation. Appendix A provides more examples comparing specifically with existing other types of ein*-notation. Due to space constraints, we added these examples only in the appendix.

---

### Official Review · Reviewer_PoRD · 2025-11-02

**Soundness:** 3
**Presentation:** 3
**Contribution:** 3
**Rating:** 6
**Confidence:** 3

**Summary:**

This paper proposes a new notation for tensor operations called einx. Previous attempts, such as einsum and its variants, have been proposed and widely used in the machine learning community. However, these notations have limitations in expressive power by definition, and their semantics are not always straightforward to understand. In this paper, the authors propose einx, a new notation focusing on vectorization. In the proposed notation, an einx expression can be derived by rewriting an implementation originally written using for-loops. The proposed notation enables concise descriptions of a wide range of tensor operations.

**Strengths:**

The strengths of this paper are as follows:

## Convenience of a unified notation
As the authors claim, the proposed notation enables a unified expression of various tensor operations. In particular, the ability to represent concatenated axes, such as in `einx.id`, is interesting. Specifically, in the expression of attention in Sec 5.2. 5.2, the use of `einx` indeed allows the attention mechanism to be written in a small number of lines.

## Beginner-friendly for elementary examples
In the elementary procedure of this method, one first considers a standard loop-based formulation and then converts it into `einx` notation by arranging the corresponding ids. This step-by-step approach seems intuitive and accessible even to beginners.

## Introduction and related work
I appreciate Sections 1 and 2 of this paper. The authors provide a summary of the history of past `ein*` notations, which should serve as valuable material for future researchers considering similar notational systems.

**Weaknesses:**

On the other hand, the weaknesses of this paper are as follows:

##  Mismatch with the venue due to a lack of quantitative evaluation
While the proposed notation is interesting, there is no quantitative evaluation demonstrating its actual effectiveness. Therefore, it isn't easy to judge whether the proposed method is scientifically sound as a research contribution. Since quantitative evaluation is not necessarily required for notation proposals, this work would be more suitable for venues such as MLOSS or the ACMMM Open Source Competition rather than the ICLR main conference.

## Still complex for complex examples
Although the proposed notation is easy to understand in elementary examples, it remains complex for complicated cases. For instance, an expression like `z = einx.multiply("a b c, b -> c b a", x, y)` is understandable by reasoning from the corresponding loops. Still, the attention example requires considerable familiarity to interpret. Therefore, the effectiveness of this notation (i.e., whether it truly makes complex operations easier to understand) depends heavily on the user's level of expertise with the notation.

**Questions:**

In Sec. 3.22, the phrase "The sum-reduction operation `np.sum(x, axis=1)` ... is decomposable" appears, but this wording may be somewhat misleading and might benefit from rephrasing. When I first read it, I interpreted it to mean that the `sum` computation can be "decomposed" into two operations, namely `y[i] = sum(x[i, :])` and `y[i] += x[i, j]`. However, that is not the intended meaning here; instead, it means that the sum operation can be expressed in two distinct ways, correct?

This is more of a comment than a question, but for complex notations (such as `einx.dot("b q (h [c]), b k (h [c]) -> ...` in Sec. 5.2), I believe understanding would be deepened by a visualization that explains them visually. For example, a simple HTML page that can be opened in a browser, where entering an `einx` expression produces a real-time visualization.

---

> ### Author Response · Authors · 2025-11-21
> **Response to reviewer PoRD**
>
> **W1: "Mismatch with the venue"**
>
> Besides our points in [Venue placement](https://openreview.net/forum?id=QqvQ3iAdpC&noteId=brHSSrpTHY), we would like to emphasize that the core contribution is our novel notation and paradigm for expressing and thinking tensor operations, rather than the software. We include our software implementation of einx notation as a proof-of-concept and for reference, but do not focus on its functionality in the main paper (e.g., how einx expressions are comiled to Python code for a given framework). The notation and paradigm are independent of how they are implemented in a software library.
>
> **W2: "Still complex for complex examples [...] the attention example requires considerable familiarity to interpret"**
>
> einx does require familiarity with the notation to understand, but it provides some mechanisms that ease interpretability:
>
> (1) The analogy with for-loop notation allows representing any einx notation as a code snippet with for-loops. E.g., the attention example
> ```
> z = einx.dot("b q h [c], b k h [c] -> b q k h", x, y)
> ```
> leads to the following representation with for-loops (as described in Sec. 4.1):
> ```
> for b in range(...): for q in range(...): for k in range(...): for h in range(...):
>     z[b, q, k, h] = dot(x[b, q, h, :], y[b, k, h, :])
> ```
> Adding parentheses around h and c simply indicates that the corresponding axes are flattened in the tensor.
>
> (2) einx compiles operations to function calls for a given tensor framework, and allows inspecting the compiled code. Anyone familiar with Numpy-like notation may use this as a means to understand a given expression. For instance,
> ```
> einx.sum("a ([b] c)", x, c=4)
> ```
> compiles to the following code when using Jax and invoking with a tensor of shape (8, 24):
> ```
> import jax.numpy as jnp
> def op(a):
>     a = jnp.reshape(a, (8, 6, 4))
>     a = jnp.sum(a, axis=(1,))
>     return a
> ```
> We emphasize that Numpy-like notation consists of a much larger, more complex API than einx with many inconsistent interfaces for how vectorization is expressed (c.f. Appendix B). From our experience, the reason why Numpy-like notation might seem simpler is merely existing familiarity with the notation, rather than its inherent simplicity of expressiveness. Unlike Numpy, einx provides a single set of rules that applies to all tensor operations.
>
> **Q1: "the sum operation can be expressed in two distinct ways"**
>
> Thank you for pointing this out. They are indeed two distinct ways of decomposing the operation. We agree that the formulation is ambiguous and have improved it for the camera-ready version.
>
> **Q2: "understanding would be deepened by a visualization"**
>
> Thank you for your suggestion. einx does contain two mechanisms to interpret/"visualize" an expression: The analogy with for-loops, and inspecting the code that einx compiles for an operation. The code may be queried by simply adding `graph=True` to an operation, like so:
> ```
> code = einx.sum("a ([b] c)", x, c=4, graph=True)
> ```
> This returns the following string:
> ```
> import jax.numpy as jnp
> def op(a):
>     a = jnp.reshape(a, (8, 6, 4))
>     a = jnp.sum(a, axis=(1,))
>     return a
> ```
> Appendix E contains many more examples. To ease understanding, we might add another backend in the future that compiles einx expressions to functions in for loop notation: While this might not be computationally efficient, it would provide an additional visualization for expressions.

---

> > ### Comment · Reviewer_PoRD · 2025-11-26
> >
> > I am satisfied with the authors' rebuttal and will maintain my current positive score. The `graph=True` option is very intuitive, and implementing it would definitely help beginners.

---

### Author Response · Authors · 2025-11-20
**Answer to all reviewers**

We thank all reviewers for their constructive feedback. We will address the four main points raised by the reviews in this comment and answer individual questions in the comments to each review.
# Novelty
Several reviews raised questions about the novelty of our proposed notation w.r.t. einsum/einops and our observations on the role of vectorization in tensor operations. We emphasize that einx is not an extension of einsum/einops, but follows a fundamentally different paradigm of tensor programming.

Our observations on vectorization (which the design of einx is based on) are not recognized by or incorporated into existing Numpy-like or ein* notations:
* Numpy (Harris et al.) refers only to some element-wise operations (e.g., np.add) as "vectorized", but does not recognize that most other operations (e.g., np.sum, np.tensordot, np.linalg.solve) also inherently perform vectorization. This is also evident by the fact that these operations follow different rules for how their inherent vectorization is expressed (see Appendix B). In contrast, we subsume the different interfaces under a single notation, demonstrating that "it's all just vectorization".
* einsum/einops do not recognize the role of vectorization in tensor operations and contain design choices that distinctly contradict these observations (see Sec. 5.1):
	- Lack of distinction between vectorized axes and sub-tensor argument axes (i.e., no brackets).
	- No analogy of expressions with loop notation.
	- Functions `rearrange` and `repeat` compute the same elementary operation.
	- The naming of functions is not related to the underlying elementary operations: `einsum` is not called `dot`, `rearrange` and `repeat` are not called `identity`. It instead follows an unrelated principle: "[W]e made an explicit choice to separate scenarios of “adding dimensions” (`repeat`), “removing dimensions” (`reduce`) and “keeping number of elements the same” (`rearrange`)" (Rogozhnikov et al. 2022a)
	- `repeat` and `reduce` are framed as symmetrical, despite being not.

While einsum/einops support few, particular tensor operations (see Tab. 2), einx is a universal notation and allows invoking *any* tensor operation.
# einx notation is universal
Several reviews raised questions about whether einx notation is truly universal. We want to emphasize that *any* operation can be invoked with einx syntax, even custom ones, and demonstrate this below. As an example, the einx call
```
z = einx.SOME_OPERATION("a [b], [b] c -> a c", x, y)
```
will yield the same output as
```
for a in range(...): for c in range(...):
    z[a, c] = SOME_OPERATION(x[a, :], y[:, c])
```
regardless of what is computed in `SOME_OPERATION`. The notation is universal because it applies to and supports *any* tensor operation (i.e., any `SOME_OPERATION`) and *any* vectorization by analogy with loop notation.

einx contains functions for commonly used elementary operations (e.g., einx.sum) and adapters that allow invoking custom elementary operations. For instance, the code
```
def myfunc(x, y):
    return 2 * x + torch.sum(y)
einmyfunc = einx.torch.adapt_with_vmap(myfunc)
```
defines a new einx operation which can be invoked with einx notation
```
z = einmyfunc("a [c], b [c] -> a b [c]", x, y)
```
yielding the same output as
```
for a in range(...): for b in range(...):
    z[a, b, :] = myfunc(x[a, :], y[b, :])
```
# Adoption of einx
Several reviews raise questions about the usefulness, impact and soundness of einx. While it is difficult to quantify the impact directly, the existing adoption of einx in the community (it has been publicly available for ~2 years) does provide evidence of its usefulness and impact on machine learning research. Due to the double blind review process we cannot provide references, but only state the numbers here:
- Over 900 repositories on Github list einx as a direct or indirect dependency.
- einx has been downloaded from PyPI over 10,000 times per day over the last weeks (although this includes automatic downloads, e.g., from CI).
# Venue placement
We believe that ICLR is the best fit for our paper:
- einx fundamentally changes how machine learning models are mentalized and code is written, and has already found adoption across many machine learning projects. We believe most practitioners are familiar with the problems in tensor notation that einx addresses (see Appendix B). We aim specifically at machine learning use cases by supporting common deep learning frameworks (e.g., PyTorch, Jax) and operations (e.g., softmax).
- The paper introducing einops was published at ICLR in 2022. It is similar to ours in that it provides (1) notation and (2) software. The final acceptance decision for its review (https://openreview.net/forum?id=oapKSVM2bcj) supports our view and states that the paper is "about design, not about models or algorithms" and that ICLR "expose[s] researchers and practitioners in machine learning to ideas and techniques that may advance their research and practice".

---

### Author Response · Authors · 2025-12-03
**Final comment by authors**

We thank all reviewers for their constructive feedback and discussion. We have incorporated many suggestions into our paper, and have adapted several parts to improve clarity. These include:

- Improve clarity on the universal nature of einx notation: In *4.1 Notation > Vectorization* we have reworked the text to better illustrate the definition of einx notation by analogy with loop notation, and use an arbitrary operation (named `SOME_OPERATION`) rather than concrete operations to highlight the universal applicability. In *4.2 Characteristics > Universal* we have added an example to show how some arbitrary Python function that represents a custom tensor operation can be vectorized with einx notation, and how it relates to the respective loop notation.
- Illustrate more clearly that einx is not a generalization of einsum/einops: In *5.1 General Comparison*, we now demonstrate with several concrete points that einsum and einops do not recognize the role of vectorization in tensor operations (which is what einx is based on and what allows it to be universally applicable).
- To highlight more practical advantages of einx (which some reviewers asked for), we have added Sec. 4.3 where we illustrate advantages over other tensor notations using example operations (content was originally in the appendix).
- Many minor changes to improve clarity.

---

### Meta-Review · Area_Chair_h36L · 2025-12-17

**Summary:**

This paper proposed vectorization as a general function for transforming tensor operations, and was generally received very positively by all reviewers.

I don't see any remaining, fundamental concerns. As such, this paper is a clear accept, and potentially a candidate for an oral presentation.

**Reviewer Concerns:**

Especially juN8 raised concerns whether the "all is vectorization" viewpoint is truly scientifically novel. Despite concerns, this reviewer already indicated a raised score.

**Reviewer Scores:**

* juN8 4 to 6
* PoRD kept 6
* GyJP 6 , potential increase to 8
* bcTx 8

Overall, this makes the paper a clear accept.

---

### Decision · Program_Chairs · 2026-01-26

Accept (Oral)